# Exploring Neoadjuvant Chemotherapy, Predictive Models, Radiomic, and Pathological Markers in Breast Cancer: A Comprehensive Review

**DOI:** 10.3390/cancers15215288

**Published:** 2023-11-04

**Authors:** Basma Elsayed, Ahmed Alksas, Mohamed Shehata, Ali Mahmoud, Mona Zaky, Reham Alghandour, Khaled Abdelwahab, Mohamed Abdelkhalek, Mohammed Ghazal, Sohail Contractor, Hossam El-Din Moustafa, Ayman El-Baz

**Affiliations:** 1Biomedical Engineering Program, Faculty of Engineering, Mansoura University, Mansoura 35516, Egypt; basma.mamdouh@std.mans.edu.eg; 2Department of Bioengineering, University of Louisville, Louisville, KY 40292, USA; ammost01@louisville.edu (A.A.); mohamed.shehata@louisville.edu (M.S.); ali.mahmoud@louisville.edu (A.M.); 3Diagnostic Radiology Department, Faculty of Medicine, Mansoura University, Mansoura 35516, Egypt; mona_zaky1982@mans.edu.eg; 4Medical Oncology Department, Mansoura Oncology Center, Mansoura University, Mansoura 35516, Egypt; rehamalghandour@mans.edu.eg; 5Surgical Oncology Department, Mansoura Oncology Center, Mansoura University, Mansoura 35516, Egypt; khaled14eg@mans.edu.eg (K.A.); mabdelkhalek@mans.edu.eg (M.A.); 6Electrical, Computer, and Biomedical Engineering Department, Abu Dhabi University, Abu Dhabi 59911, United Arab Emirates; mohammed.ghazal@adu.ac.ae; 7Department of Radiology, University of Louisville, Louisville, KY 40202, USA; sohail.contractor@louisville.edu; 8Faculty of Engineering, Mansoura University, Mansoura 35516, Egypt; hossam_moustafa@mans.edu.eg

**Keywords:** breast cancer, computed tomography, mammography, magnetic resonance imaging, multi-modal imaging, neoadjuvant chemotherapy, pathological markers, predictive models, radiomic markers, treatment response

## Abstract

**Simple Summary:**

Breast cancer is considered as the most common malignancy among females, and its treatment takes many forms and types. Neoadjuvant chemotherapy (NACT), which is the treatment precedes the surgical intervention, became the preferred treatment approach for some subtypes of breast tumors. However, some patients exhibit good response to the neoadjuvant treatment, while others do not. Therefore, the proactive prediction of patients’ response to NACT is a necessity to reduce the exposure to unnecessary doses of treatment, treatment costs, and side effects. Many researchers proposed prediction models to predict patients’ response to NACT either at early stage of treatment or prior to the initiation of the first cycle. They used various radiomics, pathological, and clinical predictors and markers. This review discusses some of the researches conducted the last decade based on statistical, machine learning, or deep learning approaches.

**Abstract:**

Breast cancer retains its position as the most prevalent form of malignancy among females on a global scale. The careful selection of appropriate treatment for each patient holds paramount importance in effectively managing breast cancer. Neoadjuvant chemotherapy (NACT) plays a pivotal role in the comprehensive treatment of this disease. Administering chemotherapy before surgery, NACT becomes a powerful tool in reducing tumor size, potentially enabling fewer invasive surgical procedures and even rendering initially inoperable tumors amenable to surgery. However, a significant challenge lies in the varying responses exhibited by different patients towards NACT. To address this challenge, researchers have focused on developing prediction models that can identify those who would benefit from NACT and those who would not. Such models have the potential to reduce treatment costs and contribute to a more efficient and accurate management of breast cancer. Therefore, this review has two objectives: first, to identify the most effective radiomic markers correlated with NACT response, and second, to explore whether integrating radiomic markers extracted from radiological images with pathological markers can enhance the predictive accuracy of NACT response. This review will delve into addressing these research questions and also shed light on the emerging research direction of leveraging artificial intelligence techniques for predicting NACT response, thereby shaping the future landscape of breast cancer treatment.

## 1. Introduction

Breast cancer stands as the most prevalent malignancy among females globally, comprising 11.7% of all cancer incidences worldwide, with 2.3 million new cases and 685,000 fatalities recorded in 2020. It ranks as the fifth leading cause of cancer-related deaths on a global scale [1,2]. In the United States, projections for 2023 indicate an estimated 300,590 new cases, resulting in 43,700 deaths across both genders [3]. Among females, breast cancer contributes to 31% of all cancer instances, foreseeing 297,790 new cases and 43,170 deaths in the US during 2023 [3]. The significance of early breast cancer detection cannot be overstated, holding the potential to substantially lower mortality and morbidity rates, enhance prospects of cure and survival, and facilitate accessible and cost-effective treatment options [4].

There are two primary treatment options for breast cancer: local therapy, which comprises surgery and radiation therapy; and systemic treatment, which includes chemotherapy, endocrine (hormone) therapy, and targeted therapy [5,6]. The selection of the appropriate treatment for each patient depends on various factors, such as age, menopausal status, tumor subtype, and stage, as well as the patient’s overall health and preferences [5,7]. Moreover, systemic therapy traditionally follows surgery in the adjuvant setting, while in the neoadjuvant setting, it precedes surgery [8,9]. Neoadjuvant chemotherapy (NACT) can, in some cases, reduce tumor size, downstage the disease, minimize the extent of local surgery, and increase the likelihood of breast-conserving surgery (lumpectomy) over mastectomy [10,11]. Currently, it is considered the gold standard treatment for locally advanced breast cancer (LABC), which refers to an aggressive stage of breast cancer where the tumor size exceeds 5cm and may involve the skin or chest wall [12,13,14].

Additionally, NACT became the preferred treatment approach for HER2-positive (HER2+) and triple-negative breast cancers (TNBC), even in very early breast cancer (breast mass more than 1.5 cm), according to the American Society of Clinical Oncology (ASCO) and National Comprehensive Cancer Network (NCCN) guidelines [15,16,17,18]. In the case of HER2+ tumors, the neoadjuvant regimen often comprises the HER2-targeted therapy drugs (such as trastuzumab and pertuzumab) in addition to chemotherapy (e.g., anthracycline- or taxane-based therapies). In contrast, immunotherapy drugs (e.g., pembrolizumab) may be included in the neoadjuvant treatment of TNBC tumors [18,19]. Incorporation of these new drugs yields a paradigm shift in the management of these molecular types of breast cancer and increases the pathological complete response after neoadjuvant therapy. Neoadjuvant therapy (NAT) also has a crucial role in the guidance of the adjuvant treatment by evaluating the sensitivity of the tumor to therapy [20,21]. Moreover, the tumor response to NAT can be considered a significant prognostic factor for the likelihood of recurrence [21,22]. Given the numerous advantages of NACT treatment, this review will primarily focus on exploring its aspects.

The response to NACT can be classified into four categories: pathological Complete Response (pCR), pathological Partial Response (pPR), Progressive Disease (PD), and Stable Disease with No Response (SDNR) [10,23]. Several grading systems are used to evaluate pathological response, and the definition of pCR varies depending on these systems. For instance, while some systems assess the breast only, others consider both the breast and the Lymph Nodes (LNs). These grading systems encompass the Sinn Score, Sataloff, Miller-Payne grading system, Residual Cancer Burden (RCB), and AJCC ypTNM (8th Edition), among other systems discussed in a review by See et al. [23]. The neoadjuvant treatment setting often utilizes the same chemotherapeutic drugs as the adjuvant treatment setting, potentially resulting in comparable long-term and short-term side effects. These effects may include leukemia, infertility, osteoporosis, cardiomyopathy/heart diseases, fatigue, risk of infection, neuropathy, vomiting, nausea, cognitive impairment, and hair loss [24,25]. Therefore, the early prediction of patient responses aids in determining suitable treatment regimens, thereby reducing unnecessary toxicities, costs, and side effects related to NACT treatment. As a result, significant interest exists in developing predictive methods that utilize clinical, radiomic, pathological, and molecular markers [26]. The motivation for this review article stems from the unmet need for these prediction models. It is worth noting that some researchers such as O’Donnell et al. [27], Pesapane et al. [28], and Liang et al. [29] conducted systematic reviews and meta-analyses to compare the accuracies of the predictive models based on MRI images.

Recently, Artificial Intelligence (AI) and its sub-fields, namely machine learning (ML) and Deep Learning (DL), have assumed an increasingly pivotal role in predicting patients’ responses to treatment, while also optimizing personalized or precision medicine—a paradigm that intricately considers the individuality of each patient. This trend is underpinned by promising outcomes in detecting, diagnosing, and prognosing various cancers. Figure 1 delineates the fundamental constituents of a typical AI-driven treatment response prediction system. This system leverages medical images procured through diverse imaging modalities, encompassing both anatomical and functional perspectives, as its primary model inputs, followed by an intricate radiomics feature extraction procedure. Moreover, the prediction model can be supplemented with clinical markers (e.g., CA 125 and CA15-3), pathological indicators (such as estrogen receptor (ER), progesterone receptor (PR), Ki67 index, and HER2 expression), and treatment alternatives. The ultimate culmination of these inputs is the prediction of the patient’s response to NACT treatment, encompassing potential outcomes like pCR, pPR, PD, or SDNR.

This review discusses studies published from 2010 to 2023 that utilized ML models, DL models, or statistical methods to predict responses in breast cancer patients who underwent NACT. Some researchers predicted Recurrence-Free Survival (RFS), Event-Free Survival (EFS), Disease-Specific Survival (DSS), or Progression-Free Survival (PFS), in addition to predicting the response to NACT. It is important to note that this review exclusively focuses on the outcomes of predicting patients’ responses to NACT. Studies that constructed their models based on conventional mammography and Contrast-Enhanced Spectral Mammography (CESM) images are detailed in Section 2, with their results, findings, markers, and patient numbers summarized in Table 1. Ultrasound-based studies are outlined in Section 3 and summarized in Table 2. Research-based on Positron Emission Tomography/Computed Tomography (PET/CT), Dynamic Contrast-Enhanced Magnetic Resonance Imaging (DCE-MRI), and Multi-modal imaging are addressed in Section 4, Section 5 and Section 6, respectively, with summaries presented in Tables 3–5. Throughout this paper, abbreviations such as ACC, AUC, SEN, SPE, PPV, and NPV denote accuracy, area under the curve, sensitivity, specificity, positive predictive value, and negative predictive value, respectively, in all tables.

## 2. Mammography and Contrast-Enhanced Spectral Mammography

**Mammography** serves as the standard imaging modality for the screening and detection of breast tumors. This low-dose X-ray imaging technique is commonly employed to identify masses, asymmetries, micro-calcifications, and architectural distortions with an accuracy ranging between 85% and 90% [30,31,32]. The two primary mammogram views are the mediolateral oblique (MLO) and craniocaudal (CC) views, supplemented by additional views like actual lateral, magnification, and point compression views, which are used for further assessment of abnormalities [33,34]. Notably, a limited number of studies (as indicated in Table 1) have employed mammography or contrast-enhanced spectral mammography to predict patients’ responses to NACT.

Recent AI studies predict early response to NACT using baseline mammogram images, showcased by Shin et al. [35] and Skarping et al. [36]. Shin et al. [35] developed a multi-scale patch-net resizing CC and MLO images to three scales (1792 × 1792, 1356 × 1356, 896 × 896) for a three-level image pyramid. They extracted features using ResNet-34 with a 3 × 3 kernel, and the classifier included a sigmoid activation followed by a fully connected layer. Kernel size affected performance: Area Under the Curves (AUCs) were 0.803 (kernel 3) and 0.661 (kernel 7); multi-scale outperformed single-scale (AUC = 0.803 vs. AUC < 0.73). Moreover, Skarping et al. [36] introduced a tumor detection model that distinguishes between pCR and non-pCR. The network features dual pathways based on ResNet-18. One processes tumor image patches, while the other handles contralateral cancer-free patches. Extracted features are concatenated and fed through fully connected layers to a soft-max layer. The model achieved notable metrics: AUC of 0.71, sensitivity of 0.46, and specificity of 90%. However, the study’s limitations include a heterogeneous cohort in terms of tumor subtype and NACT period.

**Contrast-Enhanced Spectral Mammography (CESM)** is a dual-energy technique where low-energy and high-energy images, obtained after iodinated contrast administration, are subtracted to create diagnostic subtraction images [37,38]. It serves as an MRI alternative, especially for patients with contraindications like severe claustrophobia or MRI-incompatible implants (pacemakers, defibrillators, neurostimulators, cochlear implants) [39,40]. CESM shows comparable diagnostic accuracy and sensitivity to MRI in various studies [40,41,42], finding applications in breast cancer screening, diagnosis, tumor staging, monitoring, and predicting NACT response [37,38,43].

In addition to employing AI methodologies for predicting NACT responses, Xing et al. [44] utilized statistical approaches to forecast patient responses to NACT. Specifically, they employed a *t*-test statistical method to assess the significance of the reduction percentage of the CESM gray value (ΔCGV) in CC and MLO views for early response prediction. Their findings indicated that gray values of pCR were notably lower compared to non-pCR instances, revealing a significant discrepancy in ΔCGV between the two response categories after the second treatment cycle. Within the pCR group, ΔCGV was observed to be higher than in the non-pCR group, with a *p*-value of less than 0.001 and t-values of 5.430 and 3.942 for CC and MLO views, respectively. Moreover, the CC view exhibited an AUC of 0.776, a sensitivity of 75%, and a specificity of 72.15% at a cut-off value exceeding 26.41. However, it is noteworthy that relying solely on a single factor, namely the gray value, could be considered a significant limitation of their study.

In a separate study, Wang et al. [45] developed a radiomics nomogram that incorporates the radiomics score along with three features: Background Parenchymal Enhancement (BPE), Human Epidermal Growth Factor Receptor-2 status (HER-2 status), and Ki-67 index. This nomogram aims to predict the effectiveness of NACT in cancer treatment, utilizing multivariate logistic regression analysis. The study yielded an impressive AUC of 0.81. Their findings clearly highlight the radiomics model’s heightened discriminative power when compared to the pathological model. Notably, the radiomics model achieved AUCs of 0.81 and 0.55 in the validation set, outperforming the pathological model in both instances. Additionally, Mao et al. [46] employed multiple ML algorithms to examine the efficacy of radiomics features derived from both the tumor and its neighboring areas, encompassing intratumoral and peritumoral regions. They derived five distinct radiomics signatures from these regions: intratumoral, 5 mm peritumoral, 10 mm peritumoral, intratumoral + 5 mm peritumoral, and intratumoral + 10 mm peritumoral. Their analysis culminated in the finding that the radiomics features extracted from the intratumoral + 5 mm peritumoral region exhibited superior predictive performance, as evaluated by the Least Absolute Shrinkage and Selection Operator (LASSO) regression, with an AUC of 0.85, sensitivity of 0.577, and specificity of 0.909.

It is worth noting that the studies conducted by Wang and Mao [45,46] exhibited certain limitations, given their nature as single-institutional retrospective investigations reliant on relatively modest sample sizes. Such constraints undermine the broad applicability of their findings. Furthermore, the utilization of manual (as seen in [45]) and semi-automated (as observed in [46]) ROI segmentation introduces the potential for inter- and intra-observer discrepancies. Table 1 provides a comprehensive overview of these studies, encompassing mammography and CESM, detailing the research objectives, patient count, markers or predictors, outcomes, and significant discoveries.

**Table 1 cancers-15-05288-t001:** Conventional Mammography & Contrast Enhanced Spectral Mammography (CESM).

Reference	Study Aim	Number of Patients & Study Type	Markers	Results & Findings
Shin et al. [35]	To construct a multi-scale patch learning method to early predict pCR to NACT using pre-NACT mammogram images.	288 patients Training (n = 228) Test (n = 60) Study type: single-center study	Clinical: N/A Pathological: N/A Radiomics: texture and shape features extracted from CC & MLO views by ResNet-32 with a kernel size of 3. (before extraction, images were resized to 3 different scales, then fixed-size patches were extracted from them. Patch sets were created by concatenating those patches)	The prediction performance using: kernel size of 3: AUC: 0.803, SEN: 0.733, SPE: 0.767 kernel size of 7: AUC: 0.661, SEN: 0.5, SPE: 0.833 They found that when using extracted patches, the model performance was affected by kernel size. In addition, using the whole CC & MLO mammogram images outperformed ROI-based approaches.
Skarping et al. [36]	To propose a DL-based model to predict the pCR to NACT depending on baseline digital mammograms.	453 patients Training (n = 400) Validation (n = 53) Study type: single-center study based on both retrospective & prospective cohorts	Clinical: N/A Pathological: N/A Radiomics: Image patches from the tumor & corresponding position in the reference image were processed in two parallel pathways CNN (based on ResNet18), and then features from the two pathways were concatenated and processed through FC & a final soft-max layer.	Prediction accuracy of the AI model: AUC: 0.71, SEN: 0.46, SPE: 0.9 They concluded that AI has the potential to assist in clinical decision-making. However, further research is needed with refined approaches and larger data sets to explore the utility of AI in predicting patients’ responses to NACT.
Xing et al. [44]	To investigate the effect of the reduction percentage of the CESM gray value (CGV) in the early prediction of patients’ response to NACT (whether pCR or non-pCR).	111 patients Study type: single-center retrospective study	Clinical: N/A Pathological: N/A Radiomics: percentage of gray value reduction (ΔCGV) [difference between pre-NACT & after 2nd cycle] obtained from CC & MLO views on the CESM subtraction images.	Before NACT, the differences in gray values between the pCR and non-pCR were not statically significant (*p*-value > 0.05). ΔCGV after two cycles in pCR patients was higher than the non-pCR (*p*-value < 0.001). The diagnostic value of ΔCGV using: CC view: AUC: 0.776, cut-off > 26.41, SEN: 75%, SPE: 72.15% MLO view: AUC: 0.733, cut-off > 13.59, SEN: 81.25%, SPE: 51.9% They found that ΔCGV can predict response to NACT after the second cycle.
Wang et al. [45]	Developed a radiomics nomogram to predict NACT-insensitive cancers prior to treatment based on CESM.	117 patients Training (n = 97) Validation (n = 20) Study type: single-center retrospective study	Clinical: N/A Pathological: Ki-67 index and HER2 status Radiomics: background parenchymal enhancement (BPE) in addition to shape- and size-based features, first-order statistical, and texture features.	Prediction accuracy for: Pathological markers + BPE AUC: 0.55, ACC: 0.65, SEN: 0.70, SPE: 0.60 Radiomics AUC: 0.81, ACC: 0.8 SEN: 0.90, SPE: 0.70 Integrating radiomics & pathological markers AUC: 0.81, ACC: 0.80, SEN: 0.90, SPE: 0.70 They found that the radiomics score has a good predictive ability; however, adding pathological markers did not significantly improve the model performance.
Mao et al. [46]	To study the performance of intratumoral and peritumoral radiomics acquired from CESM to predict the effect of NACT preoperatively.	118 patients Training (n = 81) Validation (n = 37) Study type: single-center retrospective study	Clinical: N/A Demographic: age Pathological: T stage, molecular subtype (according to ER, PR, HER2, & ki67) Radiomics: first-order statistics (describe the voxel intensity), shape-based features, texture features (gray-level co-occurre matrix (GLCM), gray-level run-length matrix (GLRLM), gray-level size zone matrix (GLSZM)), and filters (logarithm, exponential, gradient, square, square root, LBP, & wavelet) extracted from 5 ROIs. (intratumoral, 5 mm peritumoral, 10 mm peritumoral, intratumoral + 5 mm peritumoral, and intratumoral + 10 mm peritumoral regions).	The prediction accuracy for: Pathological markers: no significant risk factors were found. Radiomics: the AUCs based on: tumoral region: 0.74 5 mm peritumoral: 0.75 10 mm peritumoral: 0.78 tumor + 5 mm peritumoral: 0.85 tumor + 10 mm peritumoral: 0.84 The prediction model based on intratumoral+5 mm peritumoral yielded AUC: 0.85, SEN: 0.577, & SPE: 0.909 They concluded that the features extracted from the intratumoral + 5 mm peritumoral regions exhibited the best performance using LASSO Regression.

## 3. **Ultrasound**

Ultrasound modality relies on detecting the reflected echoes of transmitted high-frequency sound waves [47]. Several researchers have utilized ultrasound-acquired images and data to predict patients’ response to NACT at an early stage, leveraging its accessibility and affordability. Moreover, its suitability for repeated scans during treatment is attributed to its independence from contrast agents and long scanning times [48,49,50]. Table 2 provides a summary of studies utilizing B-mode images, quantitative ultrasound parameters, and other ultrasound-based modalities for the early prediction of patients’ response to NACT. It is evident from Table 2 that the most frequently employed imaging markers in the literature to predict the NACT response encompass the spectral slope (SS), spectral intercept or 0-MHz intercept (SI), mid-band fit (MBF), average scatterer diameter (ASD), average acoustic concentration (AAC), attenuation coefficient estimate (ACE), and spacing among scatterers (SAS) [51,52]. Below, we will emphasize the most relevant factors for predicting NACT by utilizing markers extracted from ultrasound images.

Tadayyon et al. [52] utilized quantitative ultrasound (QUS) parameters derived from baseline images, as well as images captured after one week, four weeks, and eight weeks of NACT, to distinguish between responders and non-responders. Employing the k-nearest neighbor (KNN) approach, they assessed the potential of US parameters in discerning these response groups. Their investigation revealed that the most favorable outcomes were achieved through a combination of MBF, SS, and SAS, yielding accuracies ranging from 60% to 77% across different post-treatment time points. Additionally, they observed enhanced predictive accuracies, ranging from 70% to 81%, when baseline QUS parameters were amalgamated. Notably, despite conducting a prospective study, limitations were evident due to a relatively small sample size and an imbalanced dataset, consisting of 42 responders and 16 non-responders. To counteract this imbalance, they implemented random sampling with replacement.

Sadeghi-Naini et al. [48], Sannachi et al. [53], DiCenzo et al. [54], and Dasgupta et al. [55] incorporated US parameters with textural features extracted from each parametric map as non-invasive predictors of NACT response. Sadeghi-Naini et al. [48] employed Linear Discriminant Analysis (LDA) to investigate the efficacy of MBF, spectral slope, and 0-MHz intercept in predicting treatment response. They discovered that the optimal separability between the response groups was achieved by combining textural and spectral features extracted from MBF and 0-MHz intercept parametric images acquired after one week of treatment, yielding sensitivity and specificity of 100% and an AUC of 1.

Furthermore, Sannachi et al. [53] and DiCenzo et al. [54] conducted textural analyses of SS, SI, MBF, ASD, AAC, ACE, and SAS parametric maps, employing three classifiers: *k*-Nearest Neighbor (KNN), Support vector Machine (SVM), and LDA. Sannachi et al. [53] investigated the predictive capabilities of spectral and textural features in relation to treatment response at one week, four weeks, and eight weeks. Their findings demonstrated that SVM outperformed KNN and LDA, achieving AUCs of 0.71, 0.87, and 0.92 at weeks 1, 4, and 8, respectively.

In contrast, DiCenzo et al. [54] conducted a multicenter study utilizing pre-treatment spectral and textural features. Their results indicated superior prediction performance by KNN with an accuracy of 87%, sensitivity of 91%, specificity of 83%, and an AUC of 0.73. The optimal features were AAC-homogeneity, SI-energy, and SAS-energy. Other classifiers exhibited inferior performance: SVM attained an accuracy of 75.6% and an AUC of 0.725, while FLD demonstrated an accuracy of 65.9% and an AUC of 0.67.

In a prospective study conducted by Dasgupta et al. [55], five quantitative US parametric maps, twenty texture maps, and eighty higher-order texture derivatives were generated to discriminate between responders and non-responders. The authors employed three ML algorithms, namely FLD, SVM, and KNN. Notably, KNN exhibited exceptional performance, surpassing the other algorithms with an accuracy of 82%, sensitivity of 87%, specificity of 81%, and an AUC of 0.86. In contrast, the AUCs of FLD and SVM did not exceed 0.79, and their accuracies remained below 70%. The authors concluded that utilizing US texture-derivative features resulted in superior predictive performance compared to using textural features alone.

On the other hand, certain studies have integrated radiomics features with pathological features, which can be considered invasive predictors. Tadayyon et al. [56], Sannachi et al. [57], and Tadayyon et al. [58] incorporated US parameters and textural features with specific pathological features, namely estrogen receptor (ER), progesterone receptor (PR), and human epidermal growth factor receptor 2 (HER2). In 2017, Tadayyon et al. [56] conducted a prospective study and employed FLD, KNN, and SVM classifiers to analyze the predictive performance of pathological features and US parameters acquired prior to treatment from both the tumor core and its margins (at thicknesses of 3, 5, and 10 mm). The KNN classifier exhibited the highest performance with an accuracy of 88%, a sensitivity of 90%, a specificity of 79%, and an AUC of 0.81 when utilizing radiomics features for the tumor core and 5mm margins. However, the incorporation of pathological features with radiomics led to a decline in performance, as the accuracy and AUC decreased to 79% and 0.71, respectively.

Furthermore, Sannachi et al. [57] categorized patients’ responses into three groups: complete responders, partial responders, and non-responders. They employed SVM to distinguish between these groups based on pathological and radiomics features extracted at 1, 4, and 8 weeks after the initiation of NACT. Their conclusion highlighted that the combination of pathological features with mean US parameters and texture features yielded the most robust prediction results, achieving accuracies of 79% at week 1, 86% at week 4, and 83% at week 8. Conversely, the accuracies achieved using either pathological features or radiomics features alone did not surpass 60% across all time points. Their findings underscored the need for a more extensive dataset encompassing an ample number of patients for each tumor subtype to facilitate more precise cross-validation.

In 2019, Tadayyon et al. [58] compared the prediction performances of the KNN classifier and the artificial neural network (ANN) classifier during the pretreatment stage. This analysis was based on features extracted from both the tumor and its margins (within a 5 mm perimeter around the tumor). The study encompassed both 2-class classification (categorizing responders and non-responders) and 3-class classification (identifying complete, partial, and non-responders). The results unveiled the superior performance of the ANN classifier across all experiments. In the context of binary classification, the ANN achieved an average accuracy of 96% and an AUC of 0.96, while the KNN demonstrated an accuracy of 65% and an AUC of 0.67. Furthermore, the study highlighted that pathological features played a pivotal role in the 3-class classification, whereas such significance was absent in the binary classification scenario.

Some studies utilized ultrasound-based techniques, including compression or strain elastography, as well as shear-wave elastography (SWE), for the purpose of monitoring and predicting NACT response. Both of these modalities evaluate the mechanical characteristics of tissues, particularly their stiffness and elasticity. Compression elastography involves assessing tissue deformation or strain following the application of static compression through a manual maneuver using an ultrasound transducer. In contrast, shear-wave elastography (SWE) quantifies the velocity of shear waves within the tissue, which are induced by focused acoustic radiation force [14,59,60]. Fernandes and colleagues [14] employed compression elastography to calculate relative changes in strain ratio (SR) of tumors throughout the treatment course. The strain ratio (SR) exhibited a noteworthy distinction between the two response groups following 2 weeks of NACT (*p*-value < 0.01). They utilized two classifiers, KNN and Naive Bayes, for response prediction. The accuracy rates achieved by KNN at 1, 4, 8 weeks, and the preoperative scans, were 60%, 73%, 74%, and 72%, respectively. Notably, the Naive Bayes classifier outperformed its counterpart, attaining accuracies of 72%, 84%, 83%, and 84% across the four time points.

Additionally, Ma et al. [60] and Gu et al. [61] investigated the interplay between shear-wave elastography parameters and pathological features in predicting response. Ma et al. [60] introduced a multivariable linear regression model and demonstrated that the combination of the Ki-67 index with relative changes in SWE parameters (tumor stiffness) after the second cycle of NACT yielded effective predictive capability, surpassing the individual parameters. The AUCs for predicting non-responders using the Ki-67 index, the relative change of stiffness after the second cycle, and their combination were 0.84, 0.82, and 0.93%, respectively. Similarly, Gu et al. [61] concluded that integrating Ki-67 with shear-wave parameters enhanced prediction performance at the mid-treatment stage, yielding an AUC of 0.80. Moreover, they identified mass characteristic frequency ($f_{\mathrm{mass}}$) as a novel predictor capable of determining the NACT endpoint for responders.

Several studies have employed DL techniques to predict treatment response, including Byra et al. [49,62], Xie et al. [63], Jiang et al. [50], Liu et al. [64], and Gu et al. [65]. In 2020, Byra et al. [62] introduced transfer learning using Siamese CNNs for pairwise image comparison between pre-treatment and post-first/second cycle images. With model fine-tuning, the AUC was 0.828 for differentiating malignant and benign masses in US images, while without fine-tuning, the AUC reached 0.847. This led to the conclusion that the discriminatory features might not effectively predict response. In 2022, Byra et al. [49] proposed an RNN capable of processing both US images and raw RF data, using pre-trained CNNs as feature extractors. AUC was 0.81 for response prediction from pre-treatment data using CNN pre-trained on US images and 0.93 using RGB image-pre-trained CNN for data after the fourth cycle. Similarly, Xie et al. [63] constructed a dual-branch CNN where baseline and post-first cycle images were fed into separate branches, yielding better results than individual information use. Their model achieved notable metrics: sensitivity (90.67%), specificity (85.67%), and AUC (0.939). For enhanced model generalization, further validation with multi-center data is recommended.

On the other hand, some researchers have incorporated invasive predictors into their DL models. Liu et al. [64], Gu et al. [65], and Jiang et al. [50] integrated pathological markers such as ER, PR, HER2, and Ki-67 into their models. Liu et al. [64] captured the dynamic changes of tumors before and after the first or second cycle using a Siamese network. The authors concluded that relying solely on pathological features did not yield satisfactory performance (with AUC not exceeding 0.52). However, combining these features with dynamic change features resulted in impressive AUCs of 0.904 and 0.952 for the two external validation cohorts.

Furthermore, Gu et al. [65] developed a stepwise DL model to predict responses after the second and fourth cycles of NACT. They assessed the prediction performance of pathological features, radiomics features extracted by the DL model, and the combination of radiomics and pathological features at two time points (after the 2nd and 4th cycles). The results, as presented in Table 2, led to the conclusion that the proposed model effectively assists in early stepwise prediction. Nomograms have also been utilized in predicting NACT responses. Yang et al. [66] and Jiang et al. [50] developed nomograms for early assessment and prediction of treatment response. Yang et al. [66] combined pathological features with manually crafted radiomics features extracted from baseline images, images taken after the second cycle, and delta radiomics (the difference between pre- and post-treatment features). Their findings revealed that the nomogram, incorporating the Ki-67 index along with radiomics features from images captured before and after the second cycle, demonstrated the highest predictive accuracy, achieving an AUC of 0.866. In contrast, Jiang et al. [50] fused pathological features with radiomics features extracted through a DenseNet201-based CNN, in addition to manually crafted features from pre-treatment and post-treatment images. The resultant nomogram achieved an AUC of 0.94, an accuracy of 83.9%, a sensitivity of 89.33%, and a specificity of 81.37%. Their study highlighted the efficacy of DL-based nomograms in effectively predicting pCR and offering significant insights for individualized therapy.

**Table 2 cancers-15-05288-t002:** Ultrasound.

Reference	Study Aim	Number of Patients & Study Type	Markers	Results & Findings
Tadayyon et al. [52]	To evaluate the potential Quantitative Ultrasound (QUS) parameters for early prediction of LABC patients’ clinical and pathological response to NACT.	58 patients (leave-one-out cross-validation) Study Type: single-center prospective study	Clinical: N/A Pathological: N/A Radiomics: B-mode images with parametric maps of QUS parameters (SS, SI, MBF, ASD, AAC, ACE, and SAS). These parameters were acquired at different time points (pre-NACT, after 1 week, 4 weeks, and 8 weeks of NACT).	The prediction ACCs using KNN based on the combination of MBF, SS, and SAS were 60%, 77%, and 75% using images acquired at weeks 1, 4, & 8, respectively. SENs: 61%, 79%, & -. SPEs: 59%, 76%, & -. Combining the QUS parameters at each week with pre-treatment achieved ACCs of 70%, 80%, and 81%, respectively. SENs: 76%, 80%, & -. SPEs: 64%, 79%, & -. Consequently, they found that incorporating pre-NACT QUS parameters could improve the prediction performance.
Sadeghi-Naini et al. [48]	To investigate the efficacy of textural analysis of quantitative ultrasound (QUS) spectral parametric maps for the early prediction of clinical and pathological response to NACT in LABC patients.	20 LABC patients Study Type: single-center study	Clinical: N/A Pathological: N/A Radiomics: Spectral biomarkers ( mid-band fit (MBF), the spectral slope, and the corresponding 0-MHz intercept) Textural analysis performed based on GLCM (contrast, correlation, and homogeneity)	The predictive performance using LDA when combining spectral and textural markers extracted from MBF and 0-MHz intercept (AUC: 1, SEN: 100%, SPE: 100%) Other combinations of features yielded AUCs from 0.59 to 0.99, SENs (40–100%), and SPEs (47–93%). They found that combining textural & spectral biomarkers showed the best separability between responders & non-responders at early stages of NACT.
Sannachi et al. [53]	To early predict LABC patients’ clinical and pathological response to NACT by developing computational algorithms based on quantitative ultrasound (QUS) & textural analysis.	100 LABC patients (leave-one-out cross-validation) in addition to an independent test set for SVM-RBF (n = 24) Study Type: single-center study	Clinical: N/A Pathological: N/A Radiomics: QUS parameters (MBF, SS, SI, ACE, SAS, ASD, AAC, and their mean values) Textural analysis performed based on GLCM (contrast, correlation, homogeneity, and energy). In addition to changes in these parameters after weeks 1, 4, and 8.	The accuracy of the SVM-RBF model in independent validation cohort at weeks 1, 4, & 8, respectively: Validation(1): ACCs: 82%, 78%, & 88% SENs: 87%, 80%, & 87% SPEs: 50%, 67%, & 100% Validation(2): ACCs: 72%, 81%, & 93% SENs: 73%, 84%, & 93% SPEs: 50%, 67%, & 100% They conclude that SVM-RBF outperformed the other classifiers (LDA & KNN) in differentiating responders & non-responders at all time points. Also, the most relevant features in distinguishing the two groups at weeks 1 & 4 were the changes in texture features, while at week 8 the change in mean QUS parameters were more significant.
DiCenzo et al. [54]	To construct a model for the early prediction of LABC patients’ clinical-pathological response to NACT using radiomics extracted from pre-NACT quantitative ultrasound (QUS) images.	82 LABC patients (leave-one-out cross validation) Study type: multi-center prospective study	Clinical: N/A Pathological: N/A Radiomics: B-mode images with parametric maps of QUS parameters (SS, SI, MBF, ASD, AAC, ACE, and SAS). In addition to textural analysis of parametric maps using GLCM (contrast, correlation, energy, homogeneity)	Features showed statistically significant differences between responders & non-responders (*p* < 0.05) were: SS, MBF, ASD, AAC, ASD-contrast, AAC-contrast, AAC-energy, and AAC-homogeneity. The best performing ML classifier was KNN as AUC: 0.73, ACC: 87%, SEN: 91%, SPE: 83% They found that the patients’ responses can be predicted based on pre-NACT QUS radiomics with acceptable accuracy.
Dasgupta et al. [55]	To evaluate the baseline QUS higher-order texture derivatives in predicting LABC patients’ clinical-pathological responses to NACT.	100 LABC patients (leave-one-out cross validation) Study type: single-center prospective study	Clinical: N/A Pathological: N/A Radiomics: QUS parameters (MBF, SS, SI, ASD,& AAC), texture features based on GLCM (contrast, energy, correlation, & homogeneity), & QUS texture-derivatives (QUS-Tex-Tex).	The AUCs yielded by 3 ML algorithms (FLD, KNN, & SVM) using (QUS-Tex-Tex) were 0.61, 0.86 & 0.79, respectively. The best prediction performance achieved by KNN using (QUS-Tex-Tex) as AUC: 0.86, ACC: 82%, SEN: 87%, SPE: 81% They concluded that QUS texture-derivative features (AAC-CON-ENE, MBF-COR-ENE, SI-COR- ENE) can predict tumor response before the initiation of NACT.
Tadayyon et al. [56]	To priori predict LABC patients’ clinical and pathological response to NACT based on QUS parameters and texture features extracted from tumor core and margins using ML algorithms.	56 LABC patients (leave-one-out cross-validation) Study type: single-center prospective study	Clinical: N/A Pathological: molecular markers: ER, PR, & HER2 Radiomics: features extracted from tumor core and margins (3, 5, 10 mm), B-mode images with parametric maps of QUS parameters (SS, SI, MBF, ASD, AAC, & ACE), image quality features (CMR, CMCR), & textural analysis of tumor core using GLCM (contrast, correlation, energy, homogeneity)	Using KNN, ER, PR, & HER2, respectively, achieved AUCs: 0.67, 0.48, & 0.37. ACCs: 61%, 71%, & 48%. SENs: 55%, 95%, & 60%. SPEs: 79%, 0%, & 14%. Three ML algorithms were used (FLD, SVM, & KNN) and KNN showed the best prediction performance depending on the tumor core and 5 mm margin, it yielded AUC: 0.81, ACC: 88%, SEN: 90%, SPE: 79%. Combining molecular markers decreased the model performance AUC: 0.71, ACC: 79%, SEN: 86%, SPE: 57%. They found that response to NACT can be predicted using non-invasive QUS features extracted from the tumor core and margins, and combining molecular markers with QUS did not improve the prediction power.
Sannachi et al. [57]	To early predict tumor clinical and pathological response to NACT in LABC patients using molecular markers, quantitative ultrasound (QUS) parameters, and textural features extracted from baseline and after 1, 4, and 8 weeks of NACT. They can differentiate between 3 groups of responses (complete, partial, and no response).	96 LABC patients (leave-one-out cross-validation) Study type: single-center study	Clinical: N/A Pathological: molecular features (PR, ER, and HER2). Radiomics: QUS parameters (MBF, SS, SI, ACE, SAS, ASD, AAC, and their mean values) Textural analysis performed based on GLCM (contrast, correlation, homogeneity, and energy).	The accuracies of the SVM-RBF classifier to differentiate between the 3 response groups at weeks 1, 4, & 8 using the following markers were: Molecular alone: 38%, 37%, & 50% (SEN: -, SPE: -). Mean QUS + texture: 54%, 60%, & 59% (SEN: -, SPE: -). Mean QUS + texture + molecular: 79%, 86%, & 83% (SEN: -, SPE: -). They found that combining molecular features with mean QUS values and texture features improved the discrimination power between the three response groups.
Tadayyon et al. [58]	To construct an artificial neural network (ANN) model to predict patients’ clinical and pathological response & survival prior to the start of NACT based on quantitative ultrasound (QUS) imaging and molecular markers.	100 patients (they can be classified either into 2 groups: responders & non-responders or into 3 groups: pCR, pPR, & no-response) Study type: prospective study	Clinical: N/A Pathological: PR, ER, and HER2 status Radiomics: B-mode images with parametric maps of QUS parameters (SS, SI, MBF, ASD, AAC, ACE, and SAS), texture features using GLCM (contrast, correlation, energy, homogeneity), and image quality metrics that compare the statistical properties of the core ROI & the margin which is 5 mm surrounding the tumor (CMR, CMCR).	The best performance was attained using ANN when differentiating patients who showed some response (pCR + pPR) from no response patients AUC: 0.96, ACC: 96%, SEN: 93%, SPE: 98%. Using the KNN classifier to differentiate (pCR + pPR) from no-response patients led to lower predicting performance AUC: 0.67, ACC: 65%, SEN: -, SPE: -. The authors found that ANN showed good predictive performance and can be used to evaluate the effectiveness of the treatment as a step toward personalized medicine.
Fernandes et al. [14]	To evaluate the ability of ultrasound elastography to differentiate between responders (pCRs) and non-responders (non-pCR) to NACT by monitoring changes in tumor stiffness induced by treatment.	92 LABC patients (leave-one-out cross-validation) Study type: single-center study	Clinical: N/A Pathological: N/A Radiomics: The mean strain ratio (SR), its percentage decrease relative to baseline, and B-mode images registered to elastography color maps.	Using the Naive Bayes classifier, pCR was distinguished from non-pCR at weeks 1, 4, 8, and at preoperative scan, respectively, achieved: AUCs: 0.64, 0.75, 0.77, & 0.81. ACCs: 72%, 84%, 83%, & 84%. SENs: 80%, 85%, 87%, & 84%. SPEs: 64%, 83%, 80%, & 85%. Using KNN & same time points AUCs: 0.44, 0.72, 0.66, & 0.64. ACCs: 60%, 73%, 74%, & 72%. SENs: 84%, 81%, 95%, & 85%. SPEs: 36%, 65%, 54%, & 55%. Their findings include: 1. Changes in the strain ratio (SR) correlate with tumor response. 2. Strain elastography can be used to predict response after 2 weeks. 3. The best classification performance attained at the preoperative scan using the NB classifier.
Ma et al. [60]	To investigate the potential utility of share wave elastography (SWE) and Ki67 index as response predictors to NACT in invasive breast cancer.	66 patients (response was classified according to the RCB protocol to RCB 0, I, II, &III). Study type: single-center prospective study	Clinical: N/A Pathological: ER, PR, and Ki67 index Radiomics: tumor stiffness (E) at different time points: before NACT (E0), after 1st & 2nd cycles (E1,E2), and before surgery(E6), in addition to their relative changes from baseline (ΔE1, ΔE2, ΔE6).	The accuracies of predicting (pCR & RCBI) versus (RCBII & RCBIII) using ΔE2, Ki67, and their combination, respectively, yielded: AUCs: 0.76, 0.79, & 0.88. SENs: 66.7%, 66.7%, & 100%. SPEs: 88.9%, 96.3%, & 72.2&. However, the accuracies of predicting (RCB-III) versus other response groups using the same features achieved AUCs: 0.82, 0.84, & 0.93. SENs: 68.18%, 86.36%, & 95.45. SPEs: 79.55%, 72.73%, & 79.55%. They found that a multivariable linear regression model combining ki67 with SWE parameters after the 2nd cycle of NACT showed better diagnostic performance than using each of them alone.
Gu et al. [61]	To evaluate the role of share wave elastography (SWE) in early predicting of invasive breast cancer patients’ response to NACT according to the RCB score.	62 patients (leave-one-out cross-validation) Study type: single-center prospective study	Clinical: N/A Pathological: ER, PR, HER2, and Ki67. Radiomics: tumor size, mean & max elasticity, ratio of mean & max elasticity (E_mean_,E_max_), mass characteristic frequency ($f_{\mathrm{mass}}$), and change of elasticity & mass characteristic frequency.	Using SWE parameters achieved AUC: 0.75, SEN: 0.77, & SPE: 0.75 at mid-course, while adding Ki67 achieved AUC: 0.80, SEN: 0.72, & SPE: 0.73. They concluded that combining Ki67 with some SWE parameters improves the prediction performance. Moreover, $f_{\mathrm{mass}}$ can be considered to be a new response predictor & can determine the NACT endpoint.
Byra et al. [62]	To propose two transfer learning approaches to early predict patients’ response to NACT based on US images acquired before and after the first and second cycles of NACT.	39 tumors from 30 patients Study type: single-center retrospective study	Clinical: N/A Pathological: N/A Radiomics: generic neural features extracted from pre-NACT images (approach 1) and the absolute difference between the neural feature vectors extracted from pre-NACT and after 1st & 2nd cycles by Siamese models 1 & 2, respectively, (approach 2). In addition to the handcrafted morphological features. (Siamese model consists of two identical Inception-ResNet-V2 CNNs).	Without fine tuning, the models based on pre-NACT mages, Siamese1, & Siamese2 yielded: AUCs: 0.781, 0.826, & 0.847. ACCs(%): 76.9, 82.1, & 76.9. SENs(%): 78.9, 68.4, & 78.9. SPEs(%): 75.1, 95, & 75.2. With fine tuning, the models showed AUCs: 0.797, 0.802, & 0.828 ACCs(%): 79.4, 76.9, & 76.9 SENs(%): 78.9, 73.6, & 84.2 SPEs(%): 80, 80, & 70 Using morphological features yielded AUCs: 0.736–0.792, ACCs: 66.6–74.3%, SENs: 63.3–73.6%, SPEs: 65–75%. Using morphological+ neural features AUCs: 0.818–0.844, ACCs: 76.9–84.6%, SENs: 68.4–73.6%, SPEs: 80–95%. They found that: 1. Pre-NACT features can be helpful response predictors. 2. Features used to differentiate benign & malignant masses may not be efficient for response prediction. 3. Morphological features yielded to lower performance than DL extracted features.
Byra et al. [49]	To develop recurrent neural networks (RNN) that can process regular US images and raw radio-frequency (RF) data to predict patients’ response to NACT.	51 breast cancers from 39 patients (5-fold cross-validation) Study type: single-center study	Clinical: N/A Pathological: N/A Radiomics: 3 pre-trained networks were used as feature extractors: U-Net CNN (was developed to segment malignant and benign masses) based on (1) US images (2) RF data (3) ResNet50 CNN pre-trained using ImageNet dataset (RGB images). Then the response probability was calculated by the gated recurrent unit (GRU) block and the dense prediction layer	Using the pre-NACT data, the AUCs of the models which pre-trained based on US images, RF data, & ImageNet were 0.81, 0.72, & 0.71, respectively. SENs: 0.83, 0.57, & 0.69. SPEs: 0.70, 0.89, & 0.70. Using data acquired after 4th cycle AUCs: 0.91, 0.85, & 0.93. SENs: 0.9, 0.81, & 0.9. SPEs: 0.83, 0.83, & 0.87. They revealed that: 1. Pre-trained networks used for breast mass segmentation can be good feature extractors for response prediction problems. 2. Models based on b-mode images might be sufficient for accurate response prediction as RF data acquisition is considered to be difficult.
Xie et al. [63]	To early predict the LABC patients’ pathological response to NACT by developing a novel DL approach named the dual-branch convolution neural network (DBNN) based on ultrasound images acquired before and after the first cycle of NACT.	114 LABC patients Training (n = 91) Test (n = 23) Study type: single-center retrospective study	Clinical: N/A Pathological: N/A Radiomics: features were extracted from entire breast images by a dual-branch CNN (the i/p of 1st & 2nd branches were imaged before & after 1st cycle of NACT, respectively), each branch consists of 9 layers CNN then features were weighted and shared between each branch by FSS method.	The prediction results of: Combining the US image information from pre-NACT & after 1st cycle yielded AUC: 0.939, SEN: 90.67%, SPE: 85.67% Using only pre-NACT images achieved AUC: 0.73, SEN: 76%, SPE: 68.38% Using images after 1stcycle only, AUC: 0.739, SEN: 53.3%, SPE: 86.38%. They found that: Combining data from pre-NACT & after 1st cycle outperformed the models using each of them separately. DBNN achieved outstanding results in the noninvasive prediction of response.
Liu et al. [64]	To early predict pCR in HER2-positive breast cancer patients using a Siamese multi-task network (SMTN) which performs tumor segmentation of pre- and early-treatment longitudinal ultrasound images, followed by capturing the dynamic change information of the tumor.	393 HER2-positive breast cancer patients Training (n = 215) Validation (two cohorts n = 95 & 83) Study type: multi-center retrospective study	Clinical: N/A Demographic: Age Pathological: T stage, menopausal status, NACT regimen & cycles, tumor type, ER, PR, HER2, and Ki-67. Radiomics:- SMTN consists of two subnetworks: 1. automatic tumor segmentation (2 U-nets) 2. pCR prediction (captures the dynamic change of tumor).	Mean dice coefficient (DICE) of tumor segmentation in validation cohorts > 0.764 The accuracy of predicting pCR in the two validation cohorts using different models: Pathological model: AUC: 0.52 & 0.54, ACC: 50.5% & 37.3%, SEN: 65.9% & 83.3%, SPE: 38.9% & 24.6%. SMTN: AUC: 0.902 & 0.957, ACC: 86.8% & 92.2%, SEN: 86.3% & 94%, SPE: 87.4% & 90.4%. Pathological model + SMTN: AUC: 0.904 & 0.952, ACC: 83.7% & 88.6%, SEN: 88.4% & 80.7%, SPE: 78.9% & 96.4%. They found that SMTN could assist clinicians in the early adjustment of treatment regimes for non-pCR cases. Moreover, the performance of the clinical model was unsatisfactory, and integrating it with SMTN did not improve the performance.
Gu et al. [65]	To early predict patients’ pathological response to NACT based on US images acquired prior to NACT, and after the second and the fourth cycles using the proposed novel deep learning radiomics pipeline (DLRP) which consists of two deep learning models.	168 patients Training (n = 126) validation (n = 42) Study type: single-center prospective study	Clinical: N/A Pathological: ER, PR, and HER2 status. Radiomics: The DLRP consists of 2 models (DLR2, DLR4); each model consists of 2 Densenet121 backbones. DLR2 takes US images before and after 2nd cycle as input and the extracted features were concatenated to predict response after 2nd cycle, while DLR4 takes images before and after 4th cycle.	The prediction performance of: Pathological2 model: (based on ER, HER2) AUC: 0.717, SEN: 76.2%, SPE: 61.9% Pathological4 model: (PR, HER2, reduction of tumor volume) AUC: 0.825, SEN: 61.9%, SPE: 76.2% DLR2: AUC: 0.812, SEN: 90.5%, SPE: 47.6% DLR4: AUC: 0.937, SEN: 81%, SPE: 90.5%. They concluded that depending on pathological markers only is not reliable enough for response prediction, while DLRP can effectively aid in early stepwise prediction. Moreover, hybrid models (pathological + DLR) showed no improvements in the AUC.
Yang et al. [66]	To combine pathological markers with radiomics extracted from pre-treatment and early-treatment ultrasound images for developing a nomogram used in the early prediction of patients’ radiological response to NACT.	217 patients Training (n = 152) Test (n = 65) Study type: single-center retrospective study	Clinical: N/A Demographic: Age Pathological: Ki67, histological type, clinical staging, and molecular subtype. Radiomics: first-order intensity, shape, texture, and wavelet-based features extracted before and after the 2nd cycle of NACT and the difference between them.	Radiomics features (baseline images) yielded AUC: 0.725, ACC: 67.7%, SEN: 77.8%, SPE: 65.8%. Radiomics features (after 2nd cycle) yielded AUC: 0.793, ACC: 72.3%, SEN: 60.5%, SPE: 92.6%. The nomogram combining Ki67 and radiomics signature achieved AUC: 0.866, ACC: 78.5%, SEN: 85.2%, SPE: 79.8%. They found that a nomogram combining Ki67 and radiomics signature showed the best-predicting performance.
Jiang et al. [50]	To construct and validate a DL radiomics nomogram (DLRN) to predict pCR to NACT based on ultrasound images acquired before and after treatment.	592 patients Training (n = 356) external validation cohort (n = 236) Study type: retrospective study	Clinical: N/A Demographic: age Pathological: clinical N stage, histologic type, ER, PR, HER2, Ki67, & molecular subtype. Radiomics: US diameter reduction, handcrafted features: morphology, intensity, Coiflet wavelet filter, and texture features (first, second, and high order). In addition to features extracted by CNN based on DenseNet201.	Pathological model: univariate analysis illustrated that diameter reduction, PR, Ki67, & clinical N stage showed significant difference between pCR & non-pCR (*p* < 0.05). Radiomics extracted before NACT (RS1): AUC: 0.82, SEN: -, SPE: - Radiomics extracted after NACT (RS2): AUC: 0.92, SEN: -, SPE: - DLRN: AUC: 0.94, ACC: 83.90%, SEN: 89.33%, SPE: 81.37%. They found that the DLRN outperformed the pathological model, RS1, and RS2.

## 4. PET/CT

Positron Emission Tomography (PET) serves as a prevalent nuclear imaging method for evaluating the glycolytic metabolism of tumors [47,67]. This technique enables the characterization of primary tumors, determination of lymph node stages, and assessment of residual tumors following NACT [68,69]. Distinguishing malignant tumors from adjacent normal cells relies on the heightened glucose metabolism within abnormal tissues compared to normal ones [47,68]. Typically, PET scans are complemented by other imaging modalities like computed tomography (CT), magnetic resonance imaging (MRI), and mammography, subsequent to the introduction of a radioactive tracer such as ^18^F-fluorodeoxyglucose (^18^F-FDG), ^18^F-fluorothymidine (^18^F-FLT), ^18^F-Fluciclovine, and others [70,71]. ^18^F-fluorodeoxyglucose (^18^F-FDG) stands as the most commonly employed radioactive tracer in oncology [72,73]. Pertaining to PET parameters, the maximum standardized uptake value (SUV_max_) signifies the highest concentration of the tracer (typically FDG) within the region of interest (ROI) or volume of interest (VOI) [74,75]. Due to potential noise-induced variability in SUV_max_ values, additional parameters like SUV_peak_ and SUV_mean_ may be calculated. SUV_mean_ reflects the average standardized uptake value of all voxels within the voxel of interest, while SUV_peak_ represents the parameter derived from averaging the standardized uptake values within a fixed-size, small volume of interest containing the region with the highest SUV [76,77,78].

Beyond these PET parameters, the metabolic tumor volume (MTV) denotes the total count of voxels in a VOI exhibiting uptake surpassing a specific SUV threshold, and the total lesion glycolysis (TLG) emerges as the product of MTV and SUV_mean_ [76,79]. Despite the elevated costs and radiation exposure associated with PET scans, they hold promise in predicting treatment responses [47]. This section will delve into studies utilizing either FDG PET/CT or FLT PET/CT for the early prediction of NACT responses, and these studies will be summarized in Table 3. Furthermore, key factors will be underscored for predicting NACT outcomes by leveraging markers extracted from PET/CT images.

Buchbender et al. [80] and Andrade et al. [75] employed statistical methods, specifically the Mann–Whitney test, to predict patients’ responses. Their approach relied on assessing the relative change in the standardized uptake value (ΔSUV) between images acquired before treatment and those obtained after the second cycle. Additionally, they determined the optimal ΔSUV cut-off for distinguishing between pathological pCR and non-pCR cases, as well as responders and non-responders, utilizing receiver-operating curve (ROC) analysis. A statistically significant difference in ΔSUV between the two response groups was observed. Buchbender et al. [80] identified a ΔSUV of −89% for pCR cases and −51% for non-pCR cases, yielding a *p*-value of 0.003. Furthermore, the optimal ΔSUV threshold for discriminating pCR from non-pCR was determined to be −88%, resulting in a sensitivity of 75%, specificity of 100%, and accuracy of 92%. Similarly, differentiation of responders from non-responders was achieved using an optimal ΔSUV threshold, yielding a sensitivity of −66%, specificity of 88%, and accuracy of 89%, with an overall accuracy of 92%.

In contrast, Andrade et al. [75] discovered a significantly higher ΔSUV value for pCR cases (−81.58%) in comparison to non-pCR cases (−40.18%), with a *p*-value of 0.001. They determined the optimal ΔSUV threshold for distinguishing pCR from non-pCR to be −71.8%, resulting in a sensitivity of 83.3%, specificity of 78.5%, and accuracy of 62.5%. Moreover, their differentiation of responders from non-responders using an optimal ΔSUV threshold yielded a sensitivity of −59.1%, specificity of 68%, and accuracy of 75.0%, with an overall accuracy of 86.3%.

On the other hand, Koolen et al. [81] investigated changes in the maximum standardized uptake value (SUV_max_) after 2–3 weeks and 6–8 weeks of treatment administration. They calculated ΔSUV_max_ for the primary tumor, lymph nodes, a combination of both (using logistic regression), and the highest ΔSUV_max_ (either primary tumor or lymph node). Their findings revealed that in HER2+ tumors, the AUC for predicting response using ΔSUV_max_ after 3 weeks of NACT ranged from 0.61 to 0.74 (specifically, 0.61 for the primary tumor, 0.74 for axillary nodes, 0.67 for the combined approach, and 0.72 when using the region of the highest ΔSUV_max_). However, the AUCs derived from ΔSUV_max_ after 8 weeks did not exceed 0.64 across all regions. In contrast, the prediction of response in triple-negative breast cancer (TNBC) tumors proved to be more accurate. The AUCs achieved using ΔSUV_max_ after 2 weeks of NACT ranged from 0.76 to 0.84 (specifically, 0.76 for the primary tumor, 0.74 for axillary nodes, 0.84 for the combined approach, and 0.76 when using the region of the highest ΔSUV_max_). Remarkably, the AUCs yielded from using ΔSUV_max_ after 6 weeks exceeded 0.87.

Some researchers, such as Groheux et al. [82,83], Humbert et al. [84], and Luo et al. [85], have integrated pathological markers with PET/CT parameters to predict the response to NCAT. In 2013, Groheux et al. [82] utilized a *t*-test to examine the correlation between the response and SUV_max_ of the primary tumor and lymph nodes at baseline and after two cycles in HER2-positive patients. They found that neither the pathological markers nor the SUV_max_ at baseline significantly correlated with patients’ response; the associated *p*-values were greater than 0.08. However, non-pCR could be effectively predicted using ΔSUV_max_ after the second cycle of NACT, achieving an accuracy of 80%, a sensitivity of 85.7%, and a specificity of 75% with a cut-off value of less than −62%. In 2014, Groheux et al. [83] investigated the predictive value of PET parameters and pathological markers in triple-negative tumor response prediction. They demonstrated that using a ΔSUV_max_ threshold of −50% in the primary tumor achieved an accuracy of 80%, while a threshold of −42% achieved an accuracy of 74%, a sensitivity of 58%, and a specificity of 100%. Furthermore, they noted that pathological markers yielded lower prediction accuracy (with tumor grade achieving 54% and T-stage achieving 68%).

On the other hand, Humbert et al. [84] and Luo et al. [85] combined pathological markers with radiomics and employed logistic regression to predict treatment response. Humbert et al. [84] explored the value of ΔSUV_max_ after the first treatment cycle, in conjunction with various clinical and pathological markers (such as CA 15.3, ACE, CA-125 values, estrogen receptor, progesterone receptor, HER2 expression, Ki-67 index, androgen receptor, epidermal growth factor receptor (EGFR), Scarff-Bloom-Richardson (SBR) grade, and cytokeratin (CK) 5/6 tumor expression). Their findings indicated that high ΔSUV_max_ (*p* = 0.002), elevated Ki-67 (*p* = 0.016), and negative EGFR status (*p* = 0.042) demonstrated significant associations with pCR. They achieved a predictive accuracy of 75% for pCR cases using ΔSUV_max_, with a sensitivity of 74%, specificity of 76%, and an optimal ΔSUV_max_ cut-off value of −50%. Additionally, the authors identified that non-pCR cases could be predicted with an accuracy of 92% by combining positive EGFR status and ΔSUV_max_ less than −50%.

Luo et al. [85], on the other hand, integrated Ki-67 (a proliferation marker) with SUV_max_. They computed SUV_max_ values at baseline, after the second cycle, and after the fourth cycle of treatment. The AUC values for using Ki-67 alone, SUV_max_ after the second cycle, and SUV_max_ after the fourth cycle were 0.58, 0.744, and 0.791, respectively. Their analysis revealed that while Ki-67 alone did not significantly predict pCR, its combination with radiomics markers enhanced predictive power, achieving an AUC of 0.824, sensitivity of 92.31%, and specificity of 65.71%.

Cheng et al. [86], Antunovic et al. [87], and Li et al. [88] investigated the predictive potential of pathological markers and textural features for pCR prediction using PET/CT. Cheng and colleagues [86] explored changes in SUV_max_, MTV, TLG, and textural features (outlined in Table 3) between pre- and post-second cycle images, alongside pathological markers. Significant differences in relative changes of SUV_max_, entropy, and coarseness were observed between pCR and non-pCR cases in the HER2-negative group, yielding AUCs of 0.928, 0.808, and 0.8, respectively, (*p*-values < 0.037). In the HER2-positive group, Δskewness and ΔSUV_max_ moderately predicted pCR with AUCs of 0.758 and 0.747, respectively, (*p*-values: 0.026 and 0.033). The authors concluded HER2 status strongly correlated with response, favoring pCR in HER2-positive tumors. Additionally, the textural analysis predicted pCR after two NACT cycles, while ΔTLG and ΔMTV lacked predictive efficiency.

Conversely, Antunovic et al. [87] and Li et al. [88] assessed pathological and radiomics markers’ predictive power for pCR before NACT initiation. Antunovic et al. [87] proposed four logistic regression models. Model (1) included age and tumor molecular subtype, achieving an AUC of 0.71. Discrimination improved slightly by adding SUV_max_ and TLG (AUC = 0.73). Model (2) combined second-order (correlation_GLCM_) and higher-order (Coarseness_NGLDM_, GLNU_GLZLM_) textural features with age and molecular subtype, yielding an AUC of 0.72. Models (3) and (4) employed different predictor thresholds (0.5 and 0.4, respectively). Model (3) achieved an AUC of 0.70, featuring age, molecular subtype, correlation, and coarseness. Model (4) attained an AUC of 0.73, incorporating model (3) features along with ER, NACT type, Ki67, and GLNU. The authors concluded that augmenting with SUV_max_, TLG, second-order, and higher-order radiomics features did not enhance response prediction performance.

Li et al. [88] employed a Random forest (RF) model for predicting treatment response by identifying PET/CT radiomics predictors. They extracted 2210 radiomic features including Run Variance_GLRLM_, Zone variance_GLSZM_, LGLRE_GLRLM_, and Difference Average_GLCM_. These radiomic features resulted in an AUC of 0.722, sensitivity of 0.733, specificity of 0.8, and accuracy of 0.767. Incorporating these features with age enhancements improved performance, yielding an AUC of 0.73, sensitivity of 0.733, specificity of 0.867, and accuracy of 80% in the independent validation set. In contrast, the pathological model exhibited poor predictive performance, with an AUC of 0.5.

As previously mentioned, various tracers can be administered for the PET/CT scan. FLT has served as an indicator of proliferation [89], and a handful of studies have utilized FLT PET/CT scans to early predict responses to NACT. Crippa et al. [90] computed the SUV_max_ values in both the primary tumor and the predominant lymph node, along with their relative changes between baseline and post-first-cycle NACT images. They introduced a linear combination based on ΔSUV_max_ in both the tumor and predominant lymph node, yielding an impressive AUC of 0.94 and a *p*-value below 0.001. This combination effectively discriminated between patients with RCB indices of 0 & I (indicating near-complete response) and patients with RCB indices of II & III (representing moderate & extensive residual disease). The authors highlighted the potential utility of FLT PET/CT in predicting NACT response, stressing the importance of further validation in a larger population.

**Table 3 cancers-15-05288-t003:** PET/CT.

Reference	Study Aim	Number of Patients & Study Type	Markers	Results & Findings
Buchbender et al. [80]	To test the ability of FDG-PET/CT to differentiate pCR lesions from non-pCR lesions early, after the second cycle of NACT.	26 patients Study type: retrospective study	Clinical: N/A Pathological: N/A Radiomics: The absolute and relative change in the maximum standardized uptake value (ΔSUV_max_) between scans performed at baseline and after the second cycle.	The Mann–Whitney test was used to discriminate between response groups. After 2nd cycle, ΔSUV were significantly higher for pCR (−89%) than non-pCR (−51%) and the *p*-value = 0.003. The optimal threshold of ΔSUV that discriminates: pCR & non-pCR : −88% (SEN: 75%, SPE: 100%) responders & non-responders: −66% (SEN: 88%, SPE: 89%) They found that FDG-PET/CT can be a non-invasive and efficient tool to early predict pCR cases.
Andrade et al. [75]	To investigate the correlation between the relative change in the standardized uptake value (SUV) and the pathological response to NACT using FDG-PET/CT.	40 patients (with invasive ductal breast carcinomas) Study type: single-center prospective study	Clinical: N/A Pathological: N/A Radiomics: relative change in the standardized uptake value (ΔSUV) between scans performed at baseline and after the second cycle.	The Mann–Whitney test was used to discriminate between response groups. After the 2nd cycle, ΔSUV were significantly higher for pCR (−81.58%) than non-pCR (−40.18%) and the *p*-value = 0.001. The optimal threshold of ΔSUV that discriminates : pCR & non-pCR : −71.8% (SEN: 83.3%, SPE: 78.5% ) responders & non-responders: −59% (SEN: 68%, SPE: 75%) They found that ΔSUV between baseline and second cycle scans can predict patients’ responses.
Koolen et al. [81]	To assess the value of FDG PET/CT scans of the primary tumor and lymph nodes in predicting pCR to NACT taking tumor subtype into consideration.	107 patients Study type: prospective study	Clinical: N/A Pathological: N/A Radiomics: ΔSUV_max_ were calculated after 2–3 weeks and 6–8 weeks for the primary tumor, lymph nodes, combining both of them (using logistic regression), and the highest ΔSUV_max_ (either tumor or lymph node).	The AUCs ranges of predicting response using ΔSUV_max_ for the 4 regions of interest were: In HER2+ tumors after 3 weeks: 0.61–0.72 after 8 weeks: 0.42–0.64 In TNBC tumors after 2 weeks: 0.76–0.84 after 6 weeks: 0.87–0.93 (SENs: -, SPEs: -) They concluded that PET/CT could accurately predict pCR in triple-negative tumors, especially after 6 weeks of NACT. However HER2+ tumors showed a weaker association between response and the changes in SUV_max_.
Groheux et al. [82]	To examine the value of ^18^F-FDG-PET/CT in the early identification of non-pCR cases after the second cycle of NACT in HER2+ breast cancer patients.	30 HER2+ locally advanced breast cancer patients Study type: single-center prospective study	Clinical: N/A Pathological: tumor grade, ER, and axillary status Radiomics: The maximum standardized uptake values of primary tumors and lymph nodes at baseline (SUV_max1_) and after the second cycle (SUV_max2_). In addition to the relative change between both time points (ΔSUV_max_).	They used a *t*-test to find the correlation between the response and tumor grade, ER, axillary status, SUV_max1_, SUV_max2_, and ΔSUV_max_, and *p*-values were 0.5, 0.8, 0.3, 0.08, 0.0001, and 0.001, respectively. Predicting non-pCR using: SUV_max2_: cut-off = 3, SEN: 85.7%, SPE: 93.8%, ACC: 90% ΔSUV_max_: cut-off < −62%, SEN: 85.7%, SPE: 75%, ACC: 80% They concluded that pathological markers and SUV_max1_ did not show a significant correlation with the response, whereas SUV_max2_ and ΔSUV_max_ did.
Groheux et al. [83]	To assess the value of FDG PET parameters and pathological markers in the early prediction of patients’ pathological response to NACT and event-free survival (EFS) in TNBC patients.	50 TNBC patients Study type: prospective study	Clinical: N/A Pathological: T-stage, N-stage, histology type, tumor grade, and inflammatory Radiomics: The maximum standardized uptake values of primary tumors and lymph nodes at baseline (SUV_max1_) and after the second cycle (SUV_max2_). In addition to the relative change between both time points (ΔSUV_max_).	The ACCs of predicting response using tumor grade and T-stage were 54% and 68%, respectively, (SEN: -, SPE: -). Predicting response using ΔSUV_max_ in the primary tumor at a cut-off = −50%, achieved an ACC of 80% (SEN: -, SPE: -), while using a cut-off = −42% achieved ACC: 74%, SEN: 58%, SPE: 100%. The threshold of −42% was chosen because it achieved a better prediction of relapse. They revealed that pathological markers were less predictive of response compared to PET parameters which can predict response in TNBC patients.
Humbert et al. [84]	To assess the value of tumor metabolic response (acquired by FDG-PET/CT), in addition to clinical and pathological markers in the early prediction of pCR to NACT.	50 TNBC patients Study type: single-center prospective study	Clinical: CA 15.3, ACE, and CA-125 values. Pathological: ER, PR, HER2 expression, Ki-67 index, AR (AR: Androgen Receptor), EGFR, SBR grade, and cytokeratin (CK) 5/6 tumor expression. Radiomics: SUV_max_ at baseline and after the first cycle, in addition to the relative difference between them (ΔSUV_max_).	High ΔSUV_max_ (*p* = 0.002), high Ki-67 (*p* = 0.016), and negative EGFR (*p* = 0.042) showed significant association with pCR. Predicting pCR using ΔSUV_max_: cut-off = −50 %, ACC: 75%, SEN: 74%, SPE: 76%. Non-pCR could be predicted by combining +ve EGFR status and ΔSUV_max_ < −50% with an ACC of 92%, SEN: -, SPE: -. They concluded that metabolic response combined with EGFR status can help in the early prediction of response.
Luo et al. [85]	To assess the value of Ki-67 expression and FDG PET/CT in predicting pathological response to NACT in LABC patients.	361 patients Training (n = 301) Validation (n = 60) Study type: single-center prospective study	Clinical: N/A Pathological: Ki-67 index (a proliferation marker) Radiomics: SUV_max_ at baseline and after the 2nd and 4th cycle of NACT. (ΔSUV1_max_: difference between baseline and 2nd cycle scans) (ΔSUV2_max_: difference between baseline and 4th cycle scans)	The prediction accuracy of pCR: Pathological markers AUC: 0.58, SEN: -, SPE: - ΔSUV1_max_ AUC: 0.744, SEN: 68.18%, SPE: 76.32%, cut-off: −65% ΔSUV2_max_ AUC: 0.791, SEN: 100%, SPE: 51.43%, cut-off: −69% Combining radiomics and Ki-67 AUC: 0.824, SEN: 92.31%, SPE: 65.71%. They found that Ki-67 alone did not show a significant value in the prediction of pCR, whereas combining Ki-67 with ΔSUV_max_ improved the prediction power of pCR.
Cheng et al. [86]	To determine whether textural features extracted from ^18^F-FDG PET/CT images acquired before and after the second cycle of treatment can predict pCR to NACT.	61 patients with LABC Study type: single-center retrospective study	Clinical: N/A Pathological: ER, PR, HER2, and Ki-67. Radiomics: The relative changes (Δ) of SUV_max_, MTV, TLG, and textural features (entropy, coarseness, and skewness which are based on GLCM, neighborhood gray-tone difference matrix (NGTDM), and histogram, respectively) between baseline and after the second cycle.	In the HER2(-) group, ΔSUV_max_, Δentropy, and Δcoarseness showed a significant difference between pCR and non-pCR and AUCs of predicting them: 0.928, 0.808 & 0.8 -At SEN: 100%, combining ΔSUV_max_ & Δentropy achieved SPE: 96%, while combining ΔSUV_max_ & Δcoarseness achieved SPE: 100%. In the HER2(+) group, ΔSUV_max_ and Δskewness moderately predicted pCR, (AUCs: 0.747 and 0.758). They found that textural features can predict pCR in HER2+ & HER2- patients. Moreover, ΔMTV & ΔTLG were not considered to be pCR efficient predictors.
Antunovic et al. [87]	To assess the role of radiomics (extracted from FDG PET/CT) combined with pathological markers in the prediction of pCR to NACT in patients with locally advanced breast cancer.	79 patients 100 iterations of 10-fold cross-validation Study type: single-center retropective study	Clinical: N/A Demographic: Age Pathological: ki-67 index, ER, HER2 status, molecular subtype (Luminal, HER2+, Tripple Negative), & type of NACT (anthracycline and taxane, trastuzumab, paclitaxel, letrozole, etc.) Radiomics: SUV _max_, TLG, first, second, and higher order texture features. Correlation_GLCM_, Coarseness_NGLDM_, and GLNU_GLZLM_	They proposed 4 logistic regression models, and the selected features for each model were mentioned in the main paragraph. The AUCs of models 1, 2, 3, and 4 were 0.71, 0.72, 0.70, and 0.73, respectively, (SENs: -, SPEs: -). They concluded that: 1. SUV_max_ and TLG did not show a good prediction performance of pCR. 2. The discriminatory power of the model did not improve by adding second and higher-order radiomics features. 3. A larger cohort is still needed to better investigate/judge the potential predictive role of radiomics.
Li et al. [88]	Construct an automated model to specify radiomics predictors of pCR and treatment response prior to NACT based on FDG PET/CT images.	100 patients Training (n = 70, 30 times 10-fold cross-validation) Independent validation (n = 30) Study type: single-center retrospective study	Clinical: N/A Demographic: Age Pathological: ER, PR, & HER2 status Radiomics: shape features (spherical ratio, surface area, compactness, etc.), intensity features, and texture features (GLCM, GLSZM, GLRLM, GLDZM, NGTDM). In addition to features based on the wavelet and Laplacian of Gaussian (LoG) filters.	Prediction accuracy for: Pathological markers with age ACC: 0.5, SEN: -, SPE: - (poor performance) Radiomics AUC: 0.722, ACC: 0.767, SEN: 0.733, SPE: 0.8 Radiomics with age AUC: 0.73, ACC: 0.8, SEN: 0.733, SPE: 0.867 They found that the radiomics model outperformed the pathological model. Moreover, incorporating age with PET-CT radiomics showed the best performance.
Crippa et al. [90]	To examine the ability of ^18^F-3${}^{\prime }$-deoxy-3${}^{\prime }$-fluorothymidine positron emission tomography (FLT PET) in predicting breast cancer patients’ pathological response after the first cycle of NACT.	15 LABC patients Study type: prospective study	Clinical: N/A Pathological: N/A Radiomics: SUV_max_ of primary tumor (SUVT_max_) and the predominant axillary node (SUVN_max_) and their relative percentage change after the 1st cycle (ΔSUVT_max_, ΔSUVN_max_). Moreover, a linear predictive score based on ΔSUVT_max_ and ΔSUVN_max_ was proposed.	The ΔSUVT_max_ can predict (pCR+RCBI) yielding AUC: 0.91, *p* < 0.001, cut-off ≤ −52.9%, ACC: 93.3%, SEN: 83.3%, SPE: 100%, while ΔSUVN_max_ can differentiate (RCBIII) from other response groups yielding AUC: 0.77, *p* = 0.119, SEN: -, SPE: -. The linear predictive score achieved AUC: 0.94, *p* < 0.001, SEN: -, SPE: - to differentiate RCB (0 & I) from RCB (II & III). They preliminary found a potential utility of FLT PET in predicting & monitoring response to NACT. However, these results need to be validated on a large patient population.

## 5. DCE-MRI

Dynamic contrast-enhanced magnetic resonance imaging (DCE-MRI) typically necessitates the administration of a contrast agent (CA) for delivering functional (dynamic) insights into the tissues. It offers the ability to reveal details about tumor vascularity by non-invasively gauging fluctuations in contrast to agent uptake [91,92]. Additionally, DCE-MRI furnishes information about tumor morphology encompassing size, shape features, and textural heterogeneity. This imaging technique finds utility in screening high-risk women, diagnosing breast cancer, staging breast tumors, evaluating treatment efficacy, and predicting early response to NACT [91,93]. Table 4 presents a compilation of studies employing DCE-MRI data to forecast NACT response based on quantitative or semi-quantitative parameters.

**NACT Prediction using DCE-MRI Radiomics**: Several studies employed statistical methods, such as the *t*-test, Mann–Whitney test, or Kruskal–Wallis test, to assess the effectiveness of textural features in early response prediction based on pre-treatment DCE-MRI images. Ahmed et al. [94] and Teruel et al. [95] extracted 16 Gray-Level Co-occurrence Matrix (GLCM) texture features and examined their significance in distinguishing various response groups using statistical techniques. Ahmed et al. [94] observed that contrast and difference variance yielded *p*-values ranging from 0.039 to 0.048, demonstrating a significant distinction between two response groups (partial responders & non-responders) using images captured at 1 and 2 minutes after contrast administration. Furthermore, they noted elevated contrast and difference variance values in the non-responders group, suggesting that tumors with heightened heterogeneity might exhibit reduced responses to chemotherapy.

In a similar vein, Teruel et al. [95] employed the same 16 GLCM features to differentiate between stable disease, partial response, and complete response. They identified that sum variance, entropy, and difference variance displayed statistical significance in distinguishing response groups at the 2-minute post-contrast mark, achieving *p*-values of 0.044, 0.042, and 0.033, respectively. Additionally, they compared textural features between stable disease and complete response groups, finding that 8 features achieved *p*-values below 0.05. The most noteworthy features were entropy, sum variance, and angular second moment. Using these features for predicting stable disease, ROC analysis produced AUCs of 0.77, 0.742, and 0.742, respectively.

Furthermore, sum variance, sum entropy, entropy, and difference variance exhibited significant differences between the pathological minimal residual disease group and the pathological non-responder group. Sum variance and sum entropy yielded AUCs of 0.689 and 0.686 for predicting non-responders. While Ahmed et al. [94] and Teruel et al. [95] employed the same 16 GLCM features, disparities in study design and response group definitions rendered their results incomparable. Nevertheless, they concurred that the optimal time point for response prediction was 2 min post-contrast, and these texture features could aid physicians in pre-treatment prediction of patient responses.

Machine learning algorithms have also been employed for predicting patients’ responses based on radiomics features, specifically GLCM or Gray-Level Run Length Matrix (GLRLM) features extracted from DCE-MRI images, as demonstrated by Giannini et al. [96], Fan et al. [97], and Cain et al. [98].

Giannini et al. [96] developed a Computer-Aided Diagnosis (CAD) system that automates tumor segmentation, extracts GLCM and GLRLM textural features from baseline images, and subsequently predicts pathological responses. They conducted predictions at both breast levels, including pCR and non-pCR cases, and breast and axillary levels encompassing pathologic complete response with or without residual metastatic lymph nodes (pCRN and non-pCRN, respectively). Logistic regression and Bayesian classifiers were employed for predicting pCR and pCRN. The logistic regression model achieved an AUC of 0.795 for pCR prediction (SEN: 80%, SPE: 69%). In contrast, the same model achieved an AUC of 0.764 for pCRN prediction (SEN: 46%, SPE: 100%). Conversely, the Bayesian classifier yielded an accuracy, sensitivity, and specificity of 70%, 67%, and 72%, respectively, for pCR prediction, and 64%, 69%, and 61%, respectively, for pCRN prediction.

Fan et al. [97] employed an evolutionary algorithm to select optimal features from pre-treatment images, encompassing morphological, dynamic, textural (GLCM), first-order statistical, and background parenchymal (BPE) attributes. Their retrospective study encompassed two patient cohorts: the main group (n=57) and the reproducibility group (n=46). Utilizing logistic regression, they predicted treatment response and conducted leave-one-out cross-validation (LOOCV) on the main cohort, yielding an AUC of 0.910, sensitivity of 87.2%, and specificity of 90.0%. Conversely, the reproducibility cohort exhibited an AUC of 0.874, sensitivity of 78.4%, and specificity of 88.9%. Moreover, validating the main cohort features on the reproducibility cohort produced an AUC of 0.713, while reproducibility cohort features on the main cohort resulted in an AUC of 0.683. BPE features were identified as performance-enhancing, with combined lesion and BPE features outperforming their individual counterparts.

Similarly, Cain et al. [98] introduced logistic regression and SVM models for predicting pCR to NAT. Encompassing 529 radiomics features, these models incorporated tumor and surrounding tissue attributes, with 12 features selected via stepwise multilinear regression. Significant features included the tumor-based attribute “change_in_variance_of_uptake” and volumetric Fibroglandular Tissue (FGT) enhancement. Models were trained on all NAT recipients and subsequently applied to sub-populations: NACT recipients and (TN/HER2+) NAT recipients. AUC values for (TN/HER2+) patients were 0.707 and 0.705 using logistic regression and SVM, respectively. Other sub-cohorts exhibited lower prediction performance, with AUCs below 0.658. The study underscored the potential of DCE-MRI images preceding NAT initiation in early pCR prediction, particularly for (triple negative/HER2+) patients undergoing NAT.

Eom et al. [99] employed statistical methods to assess the correlation between DCE-MRI and clinicopathological features with pCR. Their findings indicated that the tumor enhancement pattern (extracted from pre-NACT images) and the shrinkage pattern (extracted from post-NACT images) exhibited associations with pCR, with corresponding *p*-values of 0.017 and 0.015. Furthermore, individuals achieving pCR were more inclined to display homogeneous enhancement and a concentric tumor shrinkage pattern compared to non-pCR patients, with odds ratios of 14.66 and 8.63, respectively.

Further, Li et al. [100] investigated the correlation between pCR and four quantitative features mentioned in Table 4 through logistic regression analysis. Their analysis encompassed the entire patient cohort, revealing individual feature AUCs that did not surpass 0.79; however, a fusion of these features resulted in an enhanced AUC of 0.81. Furthermore, the researchers categorized the data into four subgroups based on tumor subtype, classified by HR/HER2 status. The researchers concluded that the integration of the four features substantially enhanced the predictive capacity for pCR across all subtypes and the entire dataset.

Several studies have employed both DCE-MRI quantitative parameters (K^trans^, K_ep_, V_e_, V_p_, and τ_*i*_) and semi-quantitative parameters (such as wash-in, wash-out, peak enhancement, etc.) to predict treatment response. Li et al. [101], Tudorica et al. [102], and Drisis [103] utilized statistical methods for response prediction. In one study, Li et al. [101] assessed both quantitative and semi-quantitative DCE parameters to identify response predictors and distinguish between responders and non-responders. The semi-quantitative parameter, signal enhancement ratio washout volume, exhibited significant differences between the two response groups with an AUC of 0.75 and a *p*-value of 0.03. Regarding quantitative parameters, K_ep_ (estimated by three models: Tofts-Kety, extended Tofts-Kety model, and fast exchange regime model) achieved AUCs of 0.78, 0.76, and 0.73, respectively. The authors proposed K_ep_ and signal enhancement ratio washout volume as potential response predictors, but emphasized the need for confirmation through further prospective studies.

In another study, Tudorica et al. [102] employed univariate logistic regression and C statistics to predict early pCR and non-pCR. They evaluated pharmacokinetic parameters using both the Tofts model (TM) and the Shutter speed model (SSM), along with the longest tumor diameter. The percentage change in K^trans^ (TM), K^trans^ (SSM), k_ep_ (TM), and τ_*i*_ after one cycle of NACT yielded C values exceeding 0.9, indicating excellent predictive capability. Moreover, v_e_ (TM) and v_e_ (SSM), estimated after one cycle or at the midpoint, demonstrated C values between 0.8 and 0.9, indicating good predictive potential. However, the percentage changes in the longest diameter after one cycle and at the midpoint exhibited poor prediction with C values below 0.7.

Additionally, Drisis et al. [103] statistically assessed the predictive potential of maximum tumor diameter, K^trans^ and V_e_ for early identification of responders and non-responders. The dataset was stratified into three subgroups based on tumor subtype. Using the pre-treatment scan, K^trans^ achieved AUCs of 0.66 (p=0.03) and 0.78 (p=0.03) for the entire population and the triple-negative subgroup, respectively. Additionally, the baseline maximum diameter exhibited higher values in the non-response group and achieved an AUC of 0.8 for the entire dataset. For the second scan after 2 or 3 NACT cycles, K^trans^ achieved AUCs of 0.79, 0.90, and 0.81 for the entire population, TN, and HER2+ groups, respectively.

On the other hand, Thibault et al. [104] extracted 1043 textural features from each of the 13 parametric maps. They employed a ridge regression model to predict RCB index values based on each map-feature pair (where RCB=0 indicates pCR, while any other RCB index indicates non-pCR). Their results revealed that distinguishing between pCR and non-pCR using the K_trans_ (SM) map with RLM and the Gray-level non-uniformity feature achieved a sensitivity and specificity of 100% (with an AUC of 1). Discrimination based on the V_e_(SSM) map with Haralick and contrast features yielded a sensitivity of 100% and specificity of 96.7%. While validation on a larger cohort is necessary, these findings suggest that analyzing tumor heterogeneity through textural analysis of DCE parametric maps can aid in the early prediction of treatment response.

Furthermore, Lee’s group [105] assessed baseline perfusion parameters (K^trans^, K_ep_, and V_e_) for both the tumor and contralateral breast’s background parenchyma (BP_CL_). They also performed 3D histogram analysis (including skewness, mean, kurtosis, 25th, 50th, and 75th percentiles) for each perfusion parameter using logistic regression. Although individual perfusion parameters exhibited limited predictive capacity for pCR, with AUCs ranging from 0.449 to 0.683, combining these parameters enhanced prediction performance. Notably, combining the skewness of tumor K^trans^ and background parenchyma K^trans^ yielded an AUC of 0.760 (*p*-value = 0.003). The most robust prediction performance emerged from combining the skewness and 50th percentile of tumor V_e_ with V_e_ of BP_CL_, achieving an AUC of 0.807 (p=0.002).

Aside from employing statistical methods, researchers also employed ML algorithms for predicting treatment response based on images acquired either before treatment or after the first cycle of NACT. Ashraf et al. [106], Braman et al. [107], Caballo et al. [108], and Drukker et al. [109] conducted their studies using pre-treatment images, whereas Machireddy et al. [110] utilized images acquired after the initial cycle of NACT along with baseline images.

Ashraf et al. [106] employed logistic regression with leave-one-out cross-validation (LOOCV) to differentiate between pCR and non-pCR cases, based on pre-treatment images from 15 patients in the I-SPY-1 trial dataset. They incorporated morphological, kinetic features, and kinetic statistic features, comparing their outcomes with commonly used features. The proposed kinetic statistic features achieved an AUC of 0.84, while conventional features such as longest diameter, size-to-energy ratio (SER), maximum peak enhancement (MPE), and features derived from the characteristic kinetic curve achieved AUCs of 0.54, 0.64, 0.66, and 0.71, respectively. Their findings suggest that heterogeneity in kinetic statistics could enhance the predictive performance for treatment response using baseline images.

Furthermore, Machireddy and colleagues [110] employed a support vector machine (SVM) classifier to distinguish between pCR and non-pCR cases based on data acquired before and after the initial treatment cycle. They evaluated the predictive performance of GLCM, GLRLM, single-resolution, and multi-resolution fractal features from K^trans^, K_ep_, V_e_, and τ_*i*_ parametric maps. They observed that GLCM and GLRLM features tended to overfit the data, yielding markedly lower AUCs in the test set and higher AUCs in the training set. Moreover, they found that multi-resolution fractal features outperformed other features, achieving AUCs of 0.63, 0.70, 0.74, 0.80, and 0.78 for K_ep_, τ_*i*_, V_e_, K^trans^, and the concatenation of all parametric maps, respectively.

Conversely, Braman et al. [107] and Caballo et al. [108] utilized intratumoral and peritumoral features extracted from pre-treatment images to predict pCR and non-pCR cases. In their study, Braman et al. [107] included 117 patients (78 in the training set and 39 in the test set) and employed five different ML classifiers to distinguish pCR cases.

The optimal AUCs for SVM, LDA, Diagonal Linear Discriminant Analysis (DLDA), Naive Bayes (NB), and Quadratic Discriminant Analysis (QDA) were 0.72, 0.69, 0.74, 0.72, and 0.74, respectively. Furthermore, an additional experiment was conducted where patients were categorized into subgroups based on hormone receptor type: (HR+/HER2-) and (TN/HER2+). The AUCs of the classifiers are outlined in Table 4. In this second experiment, a 3-fold cross-validation approach was adopted due to the limited sample size in the two subgroups. The authors discovered that kinetic features displayed no significant disparity between the two response groups in all experiments. Intriguingly, peritumoral features, obtained from the region adjacent to the tumor, could be amalgamated with intratumoral features to enhance response prediction.

Conversely, Caballo et al. [108] devised logistic regression ML models to differentiate between pCR and non-pCR cases, utilizing baseline intra-tumoral and peri-tumoral features. The multivariate analysis achieved an AUC of 0.707 when predicting pCR using the complete dataset. Additionally, upon stratifying the dataset according to tumor molecular subtype, the model demonstrated promising outcomes with AUCs of 0.824, 0.823, 0.844, and 0.803 for luminal A, luminal B, HER2-enriched, and TN tumors, respectively. The study revealed that enhancement kinetics heterogeneity and time-dependent texture (4D texture) features held predictive value. Moreover, features extracted from both the tumor core and peritumoral region exhibited significance in predicting pCR, with predictive features from the tumor core contributing to 65% and peritumoral features to 35% of the prediction.

Furthermore, Drukker et al. [109] employed baseline radiomics features to predict pCR and lymph node status (LN status) following NACT in LN-positive patients. Linear discriminant analysis was employed to assess the classification performance of each feature individually, distinguishing between responders and non-responders. In contrast to most studies, they unveiled that features extracted from the primary tumor lacked meaningful predictive capability. Intriguingly, the features derived from LNs, especially statistics, demonstrated robust predictive performance, with an AUC of up to 0.82 for pCR prediction and up to 0.72 for LN status prediction following NACT.

In contrast to conducting a comprehensive investigation of the entire tumor for response prediction, certain researchers, such as Wu et al. [111] and El Adoui et al. [112], have adopted a strategy of partitioning the tumor into distinct subregions based on their respective response characteristics. Wu and colleagues [111] employed *K*-means clustering to segment the tumor into subregions using eigenmaps derived from principal component analysis (PCA). This process yielded three clusters or subregions within the tumor, identified by patterns of contrast agent washin and washout. Within each subregion, they extracted four Haralick texture features and leveraged the variations in these features between the baseline scan and the scan conducted after one treatment cycle to predict the tumor’s response. Their analysis encompassed textural features from enhancement maps, as well as eigenmaps both of the entire tumor and of individual subregions. Notably, their findings indicated that the most effective prediction performance was achieved by utilizing radiomics predictors derived from eigenmaps of distinct tumor subregions, resulting in an impressive AUC of 0.79, a sensitivity of 75%, and a specificity of 78%.

On a related note, El Adoui et al. [112] introduced a parametric response map (PRM) as a visual representation, employing a color map to delineate subregions of the tumor in accordance with their respective response types. This method involved a comparative analysis of tumor volumes and intensities before and after the initial NCAT cycle.

The classification of patients as positive responders under the PRM paradigm hinged on the percentage of positive responses exceeding that of negative responses, and vice versa for negative responders. Notably, their analysis demonstrated a correspondence between the proposed PRM approach and the ground truth, substantiated by *p*-values of 0.19 for positive responses and 0.45 for negative responses, as well as impressive AUC values of 0.97 and 0.96, respectively, underscoring the effectiveness of their method.

El Adoui et al. [113] constructed a CNN model trained using baseline images and the images acquired after the initial NACT cycle, along with a fusion of these images to differentiate responders and non-responders to NACT. The experiments were replicated twice: once using images without tumor segmentation, and once using images with tumor segmentation by the U-Net DL architecture. When utilizing segmented tumor images, the AUC values were 0.71 for baseline images, 0.69 for images after the first cycle, and an improved 0.74 for their combination. In contrast, the unsegmented tumor images demonstrated higher AUCs of 0.79, 0.77, and notably elevated performance of 0.91 for baseline, post-first cycle, and combined images, respectively. They revealed that the utilization of unsegmented images outperformed segmented images, suggesting the potential utility of peritumoral areas in enhancing response prediction.

In a separate study, Khanna et al. [114] explored the role of transfer learning in response prediction. Their approach involved integrating pre-trained CNNs with ML techniques for prediction tasks. Feature extraction employed ResNet-18 and ResNet-50, while classification utilized ML methods like Decision Trees, Naïve Bayesian, KNN, and SVM. The dataset was divided using two strategies: 10-fold cross-validation and a hold-out validation, with a 70% training to 30% testing ratio. The authors highlighted the efficacy of integrating Fine KNN (with K = 1) and ResNet-18, employing Mann–Whitney U test for feature selection. This combination yielded outstanding predictive performance, achieving an AUC of 1, a sensitivity of 100%, a specificity of 99.3%, and an accuracy of 99.8% when using the hold-out validation technique.

**NACT Prediction using DCE-MRI Radiomics and Pathological Markers**: Researchers also explored the combination of pathological markers and radiomics markers to predict response, either before neoadjuvant chemotherapy (pre-NACT) or at an early stage. Jimenez et al. [115] studied the role of integrating radiomics features extracted from pretreatment scans with a pathological marker which is tumor-infiltrating lymphocyte (TIL) in predicting pCR cases. They suggested that patients who achieved high levels of TIL (more than 20%) and a radiomics signature value < 0.33 were considered to be pCR. Their model achieved an AUC of 0.752, an accuracy of 83.3%, a sensitivity of 55.6%, and a specificity of 97.2%. They found that this combination of features outperformed using either radiomics features or pathological features alone.

Conversely, Golden et al. [116] performed textural analysis on kinetic maps extracted from pre and post-NACT scans. Also, morphological and pathological features were utilized. The logistic regression model’s AUC values for predicting pCR, residual lymph nodes (LNs), and residual tumor and lymph nodes, using pre-NACT radiomics features, were 0.68, 0.84, and 0.83, respectively. However, the use of pathological markers alone exhibited limited predictive power, with AUCs below 0.6 for pCR and residual tumor and LNs. Conversely, it demonstrated moderate predictive power (AUC = 0.7) for predicting residual LN metastases. Moreover, patterns of tumor response did not exhibit significant predictive power in any model (AUC < 0.6).

Jahani’s group [117] extracted two distinct categories of features from the registered DCE scans, obtained before and after the initial cycle of NACT. The first category encompassed voxel-wise tumor deformation features, which elucidate alterations in tumor size, orientation, and shape (specifically, Jacobian (which is the ratio of the volume after the first cycle of NACT to the baseline volume), Anisotropic Deformation Index (ADI), and Slab-Rod Index (SRI)). The second category comprised voxel-wise changes in kinetic features (PE, WIS, WOS, and SER). These radiomics features achieved an AUC of 0.74 when employed in a logistic regression model to predict pCR. Additionally, an exploration into demographic and pathological markers’ potential for predicting pCR and RFS was conducted. The combination of age, race, hormone receptor status, and FTV resulted in an AUC of 0.71 for predicting pCR and incorporating demographic, pathological, and voxel-wise features raised the AUC to 0.78. The authors proceeded to compare the predictive efficacy of voxel-wise features with that of aggregate features (representing mean or average value changes across the entire tumor). The aggregate features exhibited an AUC of 0.71, leading to the author’s conclusion that, for predicting pCR, voxel-wise features outperformed aggregate features.

Conversely, Sutton et al. [118] utilized a random forest classifier to evaluate and categorize pCR based on radiomics features extracted from both pre and post-NACT images, in conjunction with the tumor molecular subtype. Three models were constructed; the first model, utilizing solely radiomics features, yielded an AUC of 0.83, a sensitivity of 0.77, and a specificity of 0.69. Subsequent integration of the tumor molecular subtype with radiomics features marginally improved performance, resulting in an AUC of 0.78, a sensitivity of 0.79, and a specificity of 0.69. The third model, based on radiomics features excluding DCE-MRI intensity features, demonstrated similar performance to the first model (AUC: 0.78, SEN: 0.79, SPE: 0.69), underscoring the value of textural and edge-based radiomics analysis in pCR classification.

In a separate study, Fan and colleagues [119] scrutinized tumor heterogeneity changes via morphological, first-order statistical, and textural analysis of images obtained before and after two cycles of NACT. A Support Vector Machine (SVM) classifier was employed to predict responders and non-responders. The prediction performance of baseline images and the Jacobian map proved subpar, with AUCs of 0.568 and 0.63, respectively. Conversely, using images from the second cycle of NACT yielded an AUC of 0.767. Furthermore, a model based on feature changes between images taken during and prior to treatment resulted in an AUC of 0.726. Notably, combining features from all aforementioned images with the molecular subtype produced the most robust prediction performance, boasting an AUC of 0.809, a sensitivity of 82.6%, and a specificity of 80%.

Hussain et al. [120] integrated molecular subtyping and the Ki-67 index with radiomics features, subsequently comparing the predictive capabilities of five ML classifiers (SVM, Naive Bayes, KNN, Decision Tree, and Ensemble RUSBoosted Tree) in determining pCR patients, utilizing data from the I-SPY-1 TRIAL dataset. Employing the Ensemble RUSBoosted Tree model, when considering solely the molecular subtype, they achieved an AUC of 0.82 alongside an accuracy of 84%. Conducting a textural analysis on MRI images acquired at baseline, after the first cycle, and at mid-treatment resulted in AUCs of 0.88, 0.72, and 0.78, and corresponding accuracies of 86%, 82%, and 76%, respectively. Notably, merging the textural analysis results from the two time points (before and after the first cycle) yielded an impressive AUC of 0.96 (with accuracy at 84%). However, the integration of the molecular subtype further enhanced prediction performance, elevating the AUC to 0.98 (with accuracy reaching 94%).

Parametric response maps have been utilized to investigate regions of increased and decreased intensity within tumors at an early stage of treatment, aiding in response prediction. This approach has been adopted by Cho et al. [121] and Drisis et al. [122]. In the study by Cho et al. [121], a *t*-test was employed to compare the predictive performance of traditional pharmacokinetic parameters (specifically K^trans^, K_ep_, and V_e_) with PRM analysis. The PRM analysis involved a voxel-wise comparison between baseline images and those acquired after the first cycle of NACT. Voxels with increased intensity (>10%) were labeled as PRM_SI+_, while those with decreased signal intensity were labeled as PRM_SI-_. Contrary to the conclusions drawn by many studies, the authors found no significant difference between pCR and non-pCR groups in terms of pharmacokinetic (PK) parameters and tumor volume change. Cho and colleagues demonstrated that voxels with increased signal intensity (PRM_SI+_) achieved an AUC of 0.77, a sensitivity of 100%, and a specificity of 71% at a cut-off of 20.8%. Additionally, they examined the potential of pathological markers using the Fisher exact test and observed no significant distinction between pCR and non-pCR groups.

Drisis et al. [122], on the other hand, generated parametric response maps (PRMs) through an affine registration method, involving the subtraction of images acquired at baseline (reference image) and after the start of NACT (transformed image). Regions where voxel values showed an increase of more than 10% were classified as non-responding regions (PRMdce+) while responding regions (PRMdce-) exhibited a decrease of more than 10% in their voxel values post-therapy initiation. Voxels not falling within these subregions were categorized as (PRMstable). The study explored the potential of both pathological markers and PRMs in predicting non-pCR. The authors determined that pathological markers and PRMs achieved AUCs of 0.71 and 0.88, respectively. They also identified PRMdce+ and grade (II) tumors as significant factors for non-pCR prediction, yielding an AUC of 0.94 with a *p*-value below 0.01.

**Table 4 cancers-15-05288-t004:** DCE-MRI.

Reference	Study Aim	Number of Patients & Study Type	Markers	Results & Findings
Ahmed et al. [94]	To evaluate the efficacy of textural features extracted from DCE-MRI in predicting response to chemotherapy.	100 patients Study type: retrospective study	Clinical: N/A Pathological: N/A Radiomics: Texture features (GLCM)	Texture features that showed a significant difference between partial responders and non-responders were contrasting (*p*-value: 0.042) and showed differences in variance (*p*-value: 0.043). They concluded that texture features showed significant differences between the two response groups at 1–2 min post-contrast time points, while pre-contrast time points did not.
Teruel et al. [95]	To examine the potential of texture analysis to predict the clinical and pathological response prior to NACT in LABC patients.	58 LABC patients Study type: single-center retrospective study	Clinical: N/A Pathological: N/A Radiomics: Two-dimensional GLCM texture features.	Eight features showed a significant difference between stable disease and complete response groups (*p*-values < 0.05). The most 3 significant features for predicting stable disease were: entropy (AUC: 0.77, SEN: 0.842, SPE: 0.684), sum variance (AUC: 0.742, SEN: 0.842, SPE: 0.684), and angular second moment (AUC: 0.742, SEN: 0.895, SPE: 0.632). They found that textural analysis can assist clinicians in the response prediction prior to therapy.
Giannini et al. [96]	To construct a CAD system that performs fully automatic segmentation of the tumor and extracts its’ textural features, in addition to the prediction of pCR & pCRN to NACT prior to treatment. (pCRN: is pCR with the absence of residual metastatic lymph nodes)	44 patients Study type: single-center retrospective study	Clinical: N/A Pathological: N/A Radiomics: gray-level co-occurrence matrixes (GLCM) & gray-level run length method (GLRLM) features.	Mono-parametic model Seven individual features correlated with pCR: AUCs from 0.674 to 0.722, SENs from 46.7% to 93.3%, and SPEs from 51.7% to 93.1%. Four features were correlated with pCRN: AUCs from 0.685 to 0.747, SENs from 84.6% to 100%, SPEs from 41.9% to 61.3%. Logistic regression model for predicting pCR: AUC: 0.795, SEN: 80%, SPE: 69% for predicting pCRN: AUC: 0.764, SEN: 46%, SPE: 100% Bayesian model for predicting pCR : ACC: 70%, SEN: 67%, SPE: 72% for predicting pCRN: ACC: 64%, SEN: 69%, SPE: 61%. They found that paients’ responses can be predicted using a CAD system that automatically segments the tumor and extracts texture features, and their results need to be validated on a larger population.
Fan et al. [97]	Using quantitative analysis of pretreatment DCE-MRI images to enhance the prediction of NACT response.	main cohort (n = 57), independent validation cohort (n = 46) Study type: single-center retrospective study	Clinical: N/A Pathological: N/A Radiomics: morphologic, dynamic, textural, first-order statistical, and background parenchymal enhancement features (BPE).	The prediction performance using multivariate logistic regression model and LOOCV based on: Main cohort: AUC: 0.910, SEN: 87.2%, SPE: 90.0%. Independent cohort: AUC: 0.874, SEN: 78.4%, SPE: 88.9% The selected features (based on the main cohort) achieved an AUC of 0.713 when tested on the independent cohort (SEN: -, SPE: -). Their results suggest that BPE features can be used as response predictors combined with lesion features.
Cain et al. [98]	To assess the utility of multivariate ML models in predicting pCR to NAT based on radiomics features extracted from DCE-MRI breast images.	288 patients divided equally into training and test. Study type: single-center retrospective study	Clinical: N/A Pathological: N/A Radiomics: size, shape, enhancement, and texture of tumors and the surrounding tissue or FGT.	Pediction AUCs using logistic regression model range from 0.589 to 0.707 (SEN: -, SPE: -), while using SVM model range from 0.593 to 0.705 (SEN: -, SPE: -). They found that pre-treatment breast MRI can be used in pCR prediction, especially in TN/HER2+ patients who had neoadjuvant therapy.
Eom et al. [99]	To evaluate the association between tumor pathological response and DCE-MRI features in addition to clinicopathologic factors in TNBC patients.	73 patients Study type: single-center retrospective study	Clinical: N/A Pathological: N/A Radiomics: pre-NACT: maximal size, shape, presentation of the tumor (mass or non-mass), margin, internal enhancement pattern (internal enhancement: heterogeneous/homogeneous/rim enhancement), and kinetics Post-NACT: tumor shrinkage pattern (concentric or dendritic)	The *p*-values of factors associated with pCR according to univariate analysis were 0.025, 0.037, & 0.009 for tumor shape, homogeneous enhancement, & concentric shrinkage pattern, respectively. The *p*-values of factors associated with pCR according to multivariate analysis were 0.017 and 0.015 for the enhancement pattern and shrinkage pattern, respectively. They concluded that homogeneous enhancement & concentric tumor shrinkage patterns are associated with pCR.
Li et al. [100]	To evaluate the performance of the predictive model combining multiple MRI features.	384 patients with LABC who enrolled in the I-SPY-2 trial. Patients were stratified into 4 subgroups according to tumor subtype Study type: retrospective study	Clinical: N/A Pathological: N/A Radiomics: functional tumor volume (FTV), longest diameter (LD), sphericity (SPH), and contralateral background parenchymal enhancement (BPE).	Using all patients: the AUCs of FTV, BPE, SPH, & LD were 0.77, 0.69, 0.69, & 0.79, respectively, (SEN: -, SPE: -); however, combining all features yielded an AUC of 0.81 (SEN: -, SPE: -). Moreover, combining all features while dividing patients into subgroups achieved AUCs of 0.83, 0.88, 0.83, & 0.82 for (HR+/HER2-), (HR+/HER2+), (HR-/HER2+), & (HR-/HER2-), respectively, (SEN: -, SPE: -). They concluded that combining the four features outperformed using each feature alone.
Li et al. [101]	To determine whether early changes in quantitative and semiquantitative DCE-MRI parameters acquired after the first cycle of NACT can differentiate between responders and non-responders, in addition to the possibility of using them as a prognostic indicator of patients’ responses to NACT.	28 patients Study type: single-center prospective study	Clinical: N/A Pathological: N/A Radiomics: Quantitative parameters (K^trans^, V_e_, K_ep_, V_p_, and τ_*i*_). Semiquantitative parameters (longest dimension, tumor volume, initial area under the curve, and signal enhancement ratio related parameters(SER)).	The AUC of SER washout volume was 0.75 and *p*-value = 0.03 (SEN: -, SPE: -) The AUCs of K_ep_ estimated by three models (Tofts-Kety, extended Tofts-Kety model, and fast exchange regime model) were 0.78, 0.76, and 0.73, and the *p*-values were 0.01, 0.02, and 0.048, respectively, (SEN: -, SPE: -). They found that the SER washout volume and K_ep_ can be used in the response prediction after one cycle of NACT.
Tudorica et al. [102]	To compare the efficacy of quantitative DCE-MRI parameters and tumor size in the early prediction of patients’ response to NACT.	28 patients Study type: N/A	Clinical: N/A Pathological: N/A Radiomics: Tumor longest diameter (LD), the percentage changes of K^trans^, K_ep_, V_e_, and τ_*i*_ after the first cycle, at the midpoint, and after NACT. PK parameters were derived by both Tofets model (TM) and the Shutter-Speed model (SSM)	Univariate logistic regression C statistics values for: Percentage changes in K^trans^, K_ep_, V_e_, and τ_*i*_ after one cycle range from 0.804 to 0.967 Changes in LD after 1st cycle and at midpoint were 0.609 and 0.673, respectively. Predicting pCR at SEN: 100% achieved SPEs: 92% & 17% using the change in K^trans^(TM) and LD, respectively. Their results suggested that LD changes were poor predictors, whereas PK parameters derived by either TM or SSM analyses were effective response predictors
Drisis et al. [103]	To determine whether pharmacokinetic (PK) parameters obtained from DCE-MRI images acquired before and after the second or third cycle of chemotherapy can predict pCR for different breast cancer subgroups.	84 LABC patients Study type: single-center retrospective study	Clinical: N/A Pathological: N/A Radiomics: Maximum diameter (D_Max_) and pharmacokinetic parameters: K^trans^, and V_e_	At baseline: K^trans^ achieved AUC: 0.66, SEN: 68%, SPE: 66% using the whole population, while it achieved AUC: 0.78, SEN: 85%, SPE: 70% using TN subgroup. At the early stage of NACT: K^trans^ achieved AUC: 0.79, SEN: 69%, SPE: 87% for the Whole population. For TN subgroup it achieved AUC: 0.9, SEN: 86%, SPE: 88%, while in HER2+ subgroup AUC: 0.81, SEN: 67%, SPE: 91%. Moreover, V_e_ achieved AUC: 0.74, SEN: 87%, SPE: 59% for the entire dataset, and AUC: 0.83, SEN: 86%, SPE: 69% for TN subgroup. They found that DCE-MRI parameters showed a significant prediction performance, especially in TN tumors.
Thibault at al. [104]	To investigate the capability of texture features generated from parametric maps of quantitative and semi-quantitative pharmacokinetic metrics acquired from DCE-MRI images at baseline and after the first cycle to early predict patients’ pathological response to NACT.	38 patients with LABC Study type: N/A	Clinical: N/A Pathological: N/A Radiomics: Texture features (GLCM or Haralick, GLRLM, GLSZM, LBP, pattern spectrum morphological operations, and basic moments) of K^trans^, V_e_, τ_*i*_, K_ep_, dK^trans^ from shutter-speed model (SSM) and standard Tofts model (SM), in addition to semi-quantitative metrics (washin, washout, SER, washin slope, and AUC).	The ridge regression model can differentiate between pCR & non-pCR using K^trans^(SM)+ GLRLM+ Gray-level nonuniformity and achieved AUC: 1, SEN: 100%, & SPE: 100%. Using V_e_(SSM)+ Haralick+ contrast feature achieved SEN: 100% and SPE: 96.7% The authors found that SSM parametric maps were more predictive than the SM parameters or the semi-quantitative metrics
Lee et al. [105]	To evaluate the utility of imaging parameters extracted from pretreatment DCE-MRI in the prediction of pCR to NACT.	74 patients Study type: single-center retrospective study	Clinical: N/A Pathological: N/A Radiomics: perfusion parameters (K^trans^, K_ep_, and V_e_) for both the tumor and the background parenchyma of contralateral breasts (BP_CL_). The 25th, 50th, and 75th percentile, skewness, mean, and kurtosis (3D histogram analysis) of each tumor perfusion parameter were obtained.	Each perfusion parameter for both tumor and BP_CL_ did not show high predictive ability for pCR (AUCs: 0.449 to 0.683, SENs: 15.4% to 100%, SPEs: 18% to 96.7%). Combination of V_e_ (BP_CL_) with 50th percentile and skewness of V_e_ in tumor had the highest predictive value (AUC: 0.807, *p* = 0.002, SEN: -, SPE: -). They revealed that the combination of perfusion parameters of tumor and BP_CL_ showed higher predictive ability for pCR than every individual parameter.
Ashraf et al. [106]	To assess the role of heterogenity-based kinetic features derived from DCE-MRI in predicting treatment pathological response.	15 patients from I-SPY-1 trial (leave-one-out cross-validation) Study type: single-center study	Clinical: N/A Pathological: N/A Radiomics: Morphological (ellipticity, circularity, perimeter, and area), kinetic features (TTP, PE, WIS, and WOS), and kinetic statistics (mean and variance of kinetic features maps).	Logistic regression model based on the proposed features (kinetic statistics) yielded an AUC of 0.84 (SEN: -, SPE: -). Most individual kinetic statistics features obtained AUCs ranging from 0.73 to 0.81 (SEN: -, SPE: -). They concluded that: Morphological features showed poor prediction performance. Moreover, the heterogeneity-based kinetic statistics outperformed the individual conventional kinetic features & their combinations, and they can be used as NACT response predictors.
Machireddy et al. [110]	To evaluate the capability of multiresolution fractal analysis of voxel-based DCE-MRI parametric maps extracted before and after the first cycle of NACT for early prediction of pCR.	55 patients Training (n = 40) Test (n = 15) Study type: single-center study	Clinical: N/A Pathological: N/A Radiomics: GLCM, GLRLM, single resolution, and multiresolution (wavelet and Fourier transformations) fractal analysis of four parametric maps (K^trans^, V_e_, K_ep_, τ_*i*_).	The AUCs of multiresolution fractal features extracted from K^trans^, K_ep_, V_e_, τ_*i*_, and all parametric maps were 0.80, 0.63, 0.74, 0.70, and 0.78, respectively. At SEN: 60%, their SPEs: 89.3%, 70.7%, 80.7%, 68.7%, and 82.7%. At SEN: 80%, their SPEs: 68.7%, 49.3%, 62%, 62%, and 62%, respectively. They revealed that multiresolution fractal features generally have better predictive performances than those extracted with conventional methods. Moreover, concatenated features from all DCE-MRI parameters improve the prediction performance rather than individual parametric maps.
Braman et al. [107]	To assess the ability of pretreatment intratumoral and peritumoral textural radiomics in predicting pCR to NACT.	117 patients Training (n = 78), Test (n = 39) Study type: retrospective study	Clinical: N/A Pathological: N/A Radiomics: pharmacokinetic parameters (PK): K^trans^, K_ep_, and V_e_. In addition to Laws energy measures, Gabor, Haralick, and CoLlAGe features. CoLlAGe: Co-occurrence of Local Anisotropic Gradient Orientations	The AUCs of LDA, DLDA, SVM, NB, and QDA classifiers (LDA: linear discriminant analysis, DLDA: diagonal linear discriminant analysis, NB: Naive Bayes, QDA: quadratic discriminant analysis), respectively, using: All patients: 0.69, 0.74, 0.72, 0.72, and 0.74 (ACC: 0.59, 0.67, 0.72, 0.64, & 0.64). (HR+,HER2-)cohort: 0.8, 0.83, 0.82, 0.81, and 0.81 (ACC: 0.77, 0.79, 0.87, 0.78, and 0.88). (TN/HER2+) cohort: 0.87, 0.89, 0.89, 0.93, and 0.85 (ACC: 0.81, 0.83, 0.82, 0.84, and 0.81). They found that intratumoral & peritumoral features can robustly predict pCR across multiple classifiers. However, PK showed no difference between pCR and non-pCR tumors.
Caballo et al. [108]	To use the spatio-temporal radiomic analysis of pretreatment DCE-MRI images in identifying patients who achieve pCR.	251 patients (leave-one-out cross validation) Study type: single-center retrospective study	Clinical: N/A Pathological: N/A Radiomics: morphology, texture temporal variation (run length, co-occurrence, & histogram-based), and enhancement kinetic heterogeneity. Features were extracted from tumoral and peritumoral regions.	Using univariate ML analysis: some individual features can be used as pCR predictors and AUCs range from 0.60 to 0.83 (SEN: -, SPE: -). Using multivariate ML logistic regression models: the AUCs of the models were 0.707, 0.824, 0.823, 0.844, & 0.803 using all patients, Luminal A, Luminal B, HER2 enriched & TN groups, respectively, (SEN: -, SPE: -). These results suggested that changes in enhancement kinetics heterogeneity and texture features over time (4D features) were significant predictors.
Drukker et al. [109]	To predict the pCR to NACT using pretreatment MRI radiomics in patients with invasive lymph node (LN positive), in addition to predicting LN status after NACT.	158 patients Study type: single-center retrospective study	Clinical: N/A Pathological: N/A Radiomics: size, shape/geometry, margin/morphology, enhancement texture, kinetics, variance kinetics, and statistics/ gray-level histogram-based features. (These features were extracted from tumor and axillary LNs)	Pre-NACT lymph node features showed significance in predicting pCR and LN status after NACT with AUCs up to 0.82 and 0.72, respectively, using the LDA classifier (SEN: -, SPE: -). They concluded that tumor features did not show significance in predicting post-NACT pCR or LN status, on the other hand, features extracted from LN did. Note: Analysis was performed using individual features, not a combination of them.
Wu et al. [111]	Using quantitative image features extracted from different tumor sub-regions to predict pathological response to NACT.	35 patients (leave-one-out cross validation) Study type: retrospictive study	Clinical: N/A Pathological: N/A Radiomics: Haralick texture features based on GLCM ( contrast, correlation, energy, and homogeneity) in three groups of predictors: enhancement map (WIS & WOS maps), eigenmaps of entire tumor region, and eigenmaps of tumor subregions.	The response prediction accuracies based on texture features extracted from the three groups of predictors were: enhancement maps: AUC: 0.67, SEN: 58%, SPE: 70% eigenmaps of the whole tumor: AUC: 0.65, SEN: 58%, SPE: 70% eigenmaps of the tumor’s subregions: AUC: 0.79, SEN: 75%, SPE: 78%. They found that eigenmaps of tumor subregions with elevated washout rate had a superior prediction performance.
El Adoui et al. [112]	To provide an algorithm based on parametric response map (PRM) which depends on the intra-tumor changes between MRI images acquired before & after the first cycle of treatment to predict patients’ response after the first cycle.	40 patients Study type: retrospective study	Clinical: N/A Pathological: N/A Radiomics: PRM creates a color map with the percentages of positive, negative, and stable tumor response after the first cycle of chemotherapy, and identifies each region with its response rate.	The AUCs were 0.97 and 0.96 for the positive response and the negative response, and the *p*-values were 0.19 and 0.45, respectively. (SEN: -, SPE: -) These results indicate the absence of a significant difference between the suggested method and the ground truth.
El Adoui et al. [113]	To conduct a CNN architecture used in predicting patients’ pathological response to NACT based on multiple DCE-MRI inputs (pre-NACT, after the first cycle of chemotherapy, and their combination).	42 LABC patients in addition to 14 external independent validation cases. Study type: retrospective study	Clinical: N/A Pathological: N/A Radiomics: features extracted from pre-NACT, after the first cycle of chemotherapy, and the combination of both scans. The model was trained using these scans with tumor segmentation and without tumor segmentation.	The performance of the model based on the following scans 1. With tumor segmentation: pre-NACT: AUC: 0.71, ACC: 69%, SEN: 81.5%, SPE: 66.2% post the first NACT cycle: AUC: 0.69, ACC: 68%, SEN: 80.3%, SPE: 66.4% both scans: AUC: 0.74, ACC: 70%, SEN: 82.4%, SPE: 68.7% 2. Without tumor segmentation: pre-NACT: AUC: 0.79, ACC: 80%, SEN: 82.1%, SPE: 67.8% post the first NACT cycle: AUC: 0.77, ACC: 80%, SEN: 81.6%, SPE: 67.1% both scans: AUC: 0.91, ACC: 88%, SEN: 92.2%, SPE: 79.1% These results underscored the superiority of combining pre- and post-NACT images in comparison to using each image type independently.
Khanna et al. [114]	To integrate pre-trained CNN with ML techniques to predict pCR to NACT using DCE-MRI images acquired prior to the initiation of treatment.	64 patients Study type: retrospective study	Clinical: N/A Pathological: N/A Radiomics: features were extracted using two pre-trained models (ResNet-18 and ResNet-50).	The best prediction performance was attained when integrating ResNet-18 (the feature extractor) with KNN (the classifier) Using hold-out validation (70:30) AUC: 1, ACC: 99.8%, SEN: 1, SPE: 99.3% Using 10-fold cross-validation AUC: 0.99, ACC: 99.1%, SEN: 99.4%, SPE: 99.1% They found that Fine KNN (K = 1) showed a superior prediction performance than other classifiers.
Jimenez et al. [115]	To predict pCR in TNBC patients who underwent neoadjuvant systemic therapy based on baseline DCE-MRI scans and tumor-infiltrating lymphocyte (TIL) levels.	80 TNBC patients (5-fold cross-validation for radiomics & radiomics+ pathological features models) Study type: single-center retrospective study	Clinical: N/A Pathological: TIL levels Radiomics: the selected features were volume, homogeneity, peak timepoint variance, uniformity, and variance.	Pathological model (TIL model): AUC: 0.632, ACC: 70%, SEN: 57.6%, SPE: 78.7%, PPV: 65.5%, NPV: 72.6% Radiomics model: AUC: 0.712, ACC: 72.5%, SEN: 85.1%, SPE: 54.6%, PPV: 72.7%, NPV: 72% Radiomics + pathological features: AUC: 0.752, ACC: 83.3%, SEN: 55.6%, SPE: 97.2%, PPV: 90.9%, NPV: 81.4% They suggested that integrating radiomics features with a pathological marker (TIL) could be utilized in predicting pCR before the initiation of the therapy.
Golden et al. [116]	To predict the patients’ response to NACT using quantitative measurements of spatial heterogeneity extracted from DCE-MRI kinetic maps.	60 patients with triple-negative early-stage breast cancer. Study type: multi-center prospective study	Clinical: N/A Demographic: Age Pathological: clinical stage, tumor grade (1, 2,and 3), ER, PR, HER2, Ki-67 index, BRCA1 status, BRCA2 status, and number of treatment cycles Radiomics: morphological features (patterns of tumor response ( such as progression, no change, regression with fragmentation, regression without fragmentation and resolution), and descriptors from BI-RADs lexicon features (such as tumor shape and margins)) and quantitative texture features (GLCM).	The AUCs of the logistic regression model using pre-NACT imaging features for predicting pCR, residual LN metastases, and residual tumor with LN metastases were 0.68, 0.84, and 0.83, respectively, (SEN: -, SPE: -). They found that pathological markers and patterns of tumor response were not significant for the prediction. Otherwise, pre-NACT radomics features yielded showed a good prediction performance.
Jahani et al. [117]	To assess changes in the intratumoral heterogeneity measured by voxel-wise image registration to perform early prediction of pCR and RFS in LABC patients.	132 LABC patients from the I-SPY-1 trial (five-fold cross-validation was performed). Study type: multi-center study	Clinical: N/A Demographic: age & race. Pathological: hormone receptor (HR) status Radiomics: functional tumor volume (FTV), voxel-wise measures of tumor deformation (Jacobian, ADI, SRI), and voxel-wise changes of parametric response maps of kinetic features (PE, WIS, WOS, SER).	The AUCs of predicting pCR using the following features were: Baseline features: (age, race, HR status, and FTV): 0.71 Voxel-wise+baseline features: 0.78 Voxel-wise features only: 0.74 Aggregate measures: (ΔPE, ΔWIS, ΔWOS, ΔSER, FTV_2_/FTV_1_): 0.71 For all features: (SEN: -, SPE: -) They found that HR status showed a significant association with pCR. Moreover, voxel-wise features showed an association with pCR, whereas the aggregate measures did not improve the model performance.
Sutton et al. [118]	To develop a classifier that assesses and classifies pCR using molecular subtypes and features extracted from pre-NACT and post-NACT scans.	273 patients 278 cancers (n = 5 bilateral) Training (n = 222 cancers) Test (n = 56 cancers) Study type: single-center retrospective study	Clinical: N/A Pathological: Molecular subtype Radiomics: first-order histogram, second-order Haralick texture, features from Gabor edge maps, Haralick texture measures computed from Gabor edge maps, and intra-tumor cluster entropy measure	The performance of the 3 RF classification models: Model (1): (radiomics only) AUC: 0.83, SEN: 0.77, SPE: 0.69 Model (2): (radiomics and molecular subtype) AUC: 0.78, SEN: 0.79, SPE: 0.69 Model (3): (radiomics without intensity metrics) AUC: 0.78, SEN: 0.79, SPE: 0.69 They suggested that radiomics features extracted before and after NACT could help in assessing pCR.
Fan et al. [119]	To demonstrate how the heterogeneity changes in DCE-MRI images at baseline & after the second cycle of NACT could affect the prediction accuracy.	114 patients Training (n = 61) Test (n = 53) Study type: retrospective study	Clinical: N/A Pathological: Molecular subtype. Radiomics: Shape, first-order statistics, GLCM, GLRLM, GLSZM, and GLDM features. (deltaRAD: means differences between pre and early NACT images).	The AUCs of the model based on: Pre-NACT radiomics: 0.568 Early-NACT radiomics: 0.767 Jacobian maps: 0.630 deltaRAD: 0.726 Combination of features: 0.771 Fusing molecular subtype with the combination of features: 0.809 The SENs of the model using the same features: 91.3%, 56.5%, 60.9%, 91.3%, 52.2%, & 82.6%, respectively, while SPEs: 36.7%, 90%, 70%, 53.3%, 96.7%, & 80%. They found that the reduction in tumor heterogeneity (indicated by texture feature) is higher among responders than non-responders.
Hussain et al. [120]	Using multiple ML classifiers to predict pCR to NACT based on molecular subtype and texture features extracted from MR images of tumor and peri-tumoral region at different treatment time points.	166 patients from I-SPY-1 trial Study type: multi-center retrospective study	Clinical: N/A Pathological: molecular subtype and Ki67 index Radiomics: Second-order texture features (extracted from GLCM of images acquired pre-, early-, and mid-treatment). In addition to peritumoral features (acquired from 3, 5, and 7-pixel morphological dilation).	The prediction performance using Ensemble Random Undersampling Boosting (RUSBoosted) classifier Tree based on: Molecular subtype AUC: 0.82, ACC: 84%, SEN: 86.48%, SPE: 76.92% Radiomics extracted from pre, early, & mid-NACT images AUCs: 0.88, 0.72, & 0.78 ACCs: 86%, 82%, & 76% SENs: 86.48%, 97.3%, & 92.85% SPEs: 84.62%, 38.46%, & 30% Combining pre and early-NACT images with molecular subtype AUC: 0.98, ACC: 94%, SEN: 94.59%, SPE: 92.31% They concluded that combining molecular subtype with radiomics extracted at pre- and early-NACT (with 3–5 pixels of the peritumoral region) had the best performance.
Cho et al. [121]	To compare the performance of the parametric response map (PRM) acquired from DCE-MRI with the pharmacokinetic parameters (PK) in the early prediction of pathological response to NACT.	48 patients Study type: prospective study	Clinical: N/A Demographic: Age Pathological: clinical stage, expression of ER, PR, HER2, and Ki-67, immunohistochemical regimen, and type of surgery. Radiomics: tumor size and volume, Parametric Response Map (PRM), and pharmacokinetic parameters (K^trans^, K_ep_, and V_e_).	The prediction performance of voxels with increased signal intensity (PRM_SI+_) in predicting pCR vs. non-pCR cutoff: 20.8%, AUC: 0.77, SEN: 100%, SPE: 71% They revealed that PRM analysis could enable the early prediction of response (after the first cycle), whereas tumor size, volume, and PK parameters do not. Moreover, pathological markers showed no differences between the pCR and non-pCR.
Drisis et al. [122]	To determine whether the parametric response mapping (PRM) can be used in the prediction of early morphological response (EMR) & pCR within 72 h after the initiation of chemotherapy.	39 patients Study type: single-center retrospective study	Clinical: N/A Demographic: Age Pathological: nodal status, tumor grade, and immunohistochemical type. Radiomics: tumor size, and Parametric Response Mapping (PRM).	Logistic regression analysis using demographic and pathological markers only obtained an AUC of 0.71, SEN: -, SPE: - PRM obtained an AUC of 0.88 for the prediction of non-pCR (SEN: -, SPE: -). Integrating the demographic and pathological markers with PRM achieved an AUC of 0.94 (SEN: -, SPE: -). They found that grade II tumors (pathological marker) and PRMdce+ (PRMdce+: voxels that showed an increment in their value of more than 10%(non-responding regions)) were significant for the prediction of non-pCR.
Comes et al. [123]	To conduct a transfer learning approach based on pre-treatment and early-treatment DCE-MRI scans to predict patients’ pathological response to NACT.	134 patients Fine-tuning (n = 108) Test (n = 26) Study type: worked on a subset of public dataset (I-SPY-1 trial)	Clinical: N/A Pathological: ER, PR, HER2, and molecular subtype. Radiomics: low-level features extracted by a pre-trained CNN from pre-and early-treatment images. Low-level features refer to local details (edges, lines, & points in the image)	Pathological features only: ACC: 69.2%, SEN: 42.9%, 78.9% Combining pathological & radiomics features: AUC: 0.90, ACC: 92.3%, SEN: 85.7%, SPE: 94.7% They concluded that low-level CNN features extracted from pre-and-early treatment images have a significant role in the early prediction of pCR.
Peng et al. [124]	To use the pretreatment DCE-MRI in comparing the prediction performances of radiomics models with DL models.	356 patients Study type: single-center retrospective study	Clinical: N/A Pathological: (molecular) ER, PR, HER2, and Ki67. Radiomics: Kinetics: (k^trans^, K_ep_, and MaxSlope) for LDA models: shape, wavelet, texture, and intensity statistical features (handcrafted features). for DL models: it can automatically learn discriminative features directly from images using ResNeXt50.	The accuracies of LDA models using: Molecular only: AUC: 0.744, ACC: 67.3%, SEN: 81.4%, SPE: 63.2% Kinetic only: AUC: 0.682, 63.8%, SEN: 68.1%, SPE: 62.5% Radiomics only: all metrics < 61% Combining all of them: AUC: 0.78, ACC: 73.1%, SEN: 79.5%, SPE: 71.2% The accuracies of DL models using: Molecular only: AUC: 0.752, ACC: 66.3%, SEN: 80.9%, SPE: 61.9% Kinetic only: AUC: 0.652, ACC: 65%, SEN: 60.8%, SPE: 66.3% Radiomics only: all metrics < 0.6 Combining all of them: AUC: 0.832, ACC: 77.2%, SEN: 78.1%, SPE: 76.9% They found that (molecular-only) models had a better prediction performance than (kinetic-only) and (radiomics-only) models. Moreover, using the DL model and combining the three feature groups outperformed the other models.
Li et al. [125]	To construct a nomogram to predict the probability of pCR in TNBC patients based on pretreatment DCE-MRI & clinicopathological features.	108 TNBC patients Training (n = 87) Validation (n = 21) Study type: single-center retrospective study	Clinical: serum CA 15-3, CA125, family history of BC, body mass index (BMI), carcinoembryonic antigen (CEA) levels. Pathological: ER, PR, HER2 status, Ki67, Lymphovascular invasion (LVI), and AR. Radiomics: TTP, tumor volume, and maximum tumor diameter.	The nomogram achieved an AUC of 0.79 in the validation cohort (SEN: -, SPE: -). They concluded that a nomogram incorporating 3 pretreatment factors (tumor volume, TTP, and AR status) had a good ability to predict pCR. Moreover, higher TTP is associated with a lower probability of achieving pCR.

Deep Learning approaches have demonstrated promising potential in predicting responses based on imaging and pathological features. Comes et al. [123], a pre-trained Convolutional Neural Network (CNN) was employed to automatically extract low-level features from images obtained before and after the initial cycle of NACT, replacing the need for manual feature extraction. The study also investigated the predictive capability of pathological features. Subsequently, the most optimal features were selected and utilized to construct an SVM classifier. The model relying on pathological features achieved an accuracy of 69.2%, a sensitivity of 42.9%, and a specificity of 78.9%. By incorporating pathological features with pre- and early-treatment radiomics features, an impressive AUC of 0.9 was achieved, along with an accuracy of 92.3%, a sensitivity of 85.7%, and a specificity of 94.7% on the independent test set.

In a study by Peng et al. [124], a comparison was made between DL and conventional ML approaches in predicting response based on baseline pathological, kinetic, and radiomics features. Conventional ML involved handcrafted radiomics feature extraction and LASSO regression for optimal feature selection, with LDA emerging as the most robust classifier among the five ML models tested. In contrast, the DL approach employed a ResNeXt50 CNN for radiomics feature extraction and a Multilayer Perceptron Neural Network (MLP) for constructing models based on pathological and kinetic features. AUC values for models utilizing radiomics, kinetics, or pathological features alone, whether through LDA or DL, did not exceed 0.752. A modest enhancement in prediction performance was observed upon combining these features. However, the DL approach, incorporating all features, notably outperformed other models with an AUC of 0.832 and an accuracy of 77.2%.

Another noteworthy contribution by Li et al. [125] involved the creation of a nomogram to predict pCR in TNBC patients. This nomogram integrated logistic regression to identify independent response predictors from a collection of pathological, clinical, and radiomics markers extracted from baseline images. Notably, the nomogram was built upon three key predictors: AR status, tumor volume, and time to peak (TTP). The predictive performance of the nomogram was impressive, achieving an AUC of 0.79 in the validation cohort. The study further highlighted that tumors exhibiting a TTP of 2 min, large volumes, and positive androgen receptor expression had a lower probability of achieving pCR.

## 6. NACT Prediction Using Multi-Modal Radiomics

Liang et al. [126] employed statistical methods, including the *t*-test and Mann–Whitney U test, to examine the efficacy of DCE-MRI parameters and ADC values acquired at baseline and after the second cycle for the early prediction of pCR. They observed that the pre-treatment parameters did not exhibit a significant difference between pCR and non-pCR, with the individual parameters’ AUCs not exceeding 0.6. Nevertheless, all parameters displayed robust predictive performance after the second NACT cycle, except V_e_ and washout. Furthermore, a combination of wash-in, TTP, and ADC yielded the highest AUC of 0.886, accompanied by a sensitivity of 87.5% and a specificity of 82.11%.

On another note, Li et al. [127] explored the potential of DCE and DWI parameters obtained before and after the initial NACT cycle, alongside the percentage change in values between the two scans, to predict the response. Through ROC curve analysis, they evaluated the predictive ability of each parameter and the derived parameter (K_ep_/ADC). The researchers uncovered that the (K_ep_/ADC) parameter, calculated after one NACT cycle, outperformed all other individual parameters, achieving an AUC of 0.88, a sensitivity of 92%, and a specificity of 78% at a threshold value of 3.32 mm^−2^ [127].

O’Flynn et al. [128] investigated the predictive ability of MRI parameters acquired before and after the second cycle of neoadjuvant chemotherapy (NACT), along with their percentage changes, to anticipate treatment response. The most significant predictors were determined using LDA. Their findings indicated that a decrease in the enhancement fraction and tumor volume signified a pCR. The enhancement fraction exhibited an AUC of 0.76, a sensitivity of 63.2%, and a specificity of 76.9% at a threshold of −7%. In comparison, the reduction in tumor volume (with a threshold of −71%) achieved an AUC, sensitivity, and specificity of 0.77, 71.4%, and 76.9%, respectively.

On the contrary, Zhao et al. [129] utilized multivariate logistic regression to explore independent predictors of pCR from radiomic features extracted from Diffusion-Weighted Imaging (DWI) and DCE-MRI scans conducted before and after the second NACT cycle. Significant pCR-independent predictors included the Apparent Diffusion Coefficient (ADC) values after 2 cycles and the percentage decrease in both the early enhanced ratio E_90_ and tumor size. The corresponding odds ratios (ORs) were calculated as 1.041 (*p* = 0.037), 0.927 (*p* = 0.004), and 0.948 (*p* = 0.011), respectively. Their prediction model yielded an impressive AUC of 0.944, a sensitivity of 100%, and a specificity of 86.7% in the validation cohort. A nomogram based on this model was subsequently established. The study also revealed that baseline MRI features, age, molecular subtype, and other pathological markers did not exhibit significant differences between the pCR and non-pCR groups.

Bian et al. [130] developed a nomogram to forecast both NACT sensitivity and pCR. They assessed the predictive capabilities of two logistic regression models based on radiomic features (as detailed in Table 5) extracted from a combination of DCE, T2WI, and DWI. Model (1) achieved AUC values of 0.56 and 0.64 for predicting NACT sensitivity and pCR, respectively. However, model (2) exhibited superior performance, achieving an AUC of 0.91, an accuracy of 81.8%, a sensitivity of 100%, and a specificity of 75% in predicting NACT sensitivity. Similarly, it achieved an AUC, accuracy, sensitivity, and specificity of 0.91, 88.9%, 88.2%, and 90.9%, respectively, in predicting pCR. The authors developed a nomogram that integrated radiomic features from both models, revealing comparable discriminative power between model (2) and the nomogram.

Some studies, such as Tahmassebi et al. [131] and Eun et al. [132], have assessed the effectiveness of radiomics features in predicting treatment response through the application of ML classifiers. In their study, Tahmassebi et al. [131] employed both qualitative and quantitative features extracted from multi-sequence MRI scans before and after the initial NACT cycle. The authors employed eight classifiers, namely LR, SVM, SGD (stochastic gradient descent), LDA, RF, DT, adaptive boosting (AdaBoost), and extreme gradient boosting (XGBoost), to predict DSS, RFS, and RCB—an indicator of whether patients achieved pCR. The results revealed XGBoost’s superior performance in terms of high accuracy and stability, with AUCs ranging from 0.8577 to 0.9430 for RCB prediction. Notably, response predictors included peritumoral edema (on T2WI), minimum ADC (on DWI), complete shrinkage pattern, changes in tumor size, and mean transit time (on DCE). Eun et al. [132] explored the potential of textural features extracted from pre- and mid-treatment images, as well as the differences between them, using contrast-enhanced T1-weighted, ADC mapping, DWI, and T2-weighted MRI. Employing seven ML classifiers, they determined that random forest surpassed other classifiers. Furthermore, mid-treatment scans demonstrated optimal prediction performance for T1 contrast-enhanced-weighted, DWI, and ADC mapping, yielding AUCs of 0.82, 0.73, and 0.69, respectively. In contrast, the difference between pre- and mid-treatment features excelled exclusively in the T2-weighted sequence, attaining an AUC of 0.67. Notably, pre-treatment feature AUCs across all sequences did not surpass 0.57.

Other researchers, including Liu et al. [133], Syed et al. [134], Chen et al. [135], Chen et al. [136], and Xiong et al. [137], explored the fusion of pathological markers with pretreatment multi-sequence MRI radiomics features to enhance the prediction of patient responses to NACT.

In a study conducted by Liu et al. [133], radiomics features were extracted from T2WI, DWI, T1+C sequences, and their combinations. Furthermore, the study assessed the predictive capacity of pathological markers. The proposed SVM models were validated across three independent cohorts (n = 99, 107, & 80). The radiomics features from individual MRI sequences yielded AUCs not exceeding 0.64 in any validation cohort. In contrast, the multi-sequence radiomics model achieved AUCs of 0.70, 0.68, and 0.79 across the respective validation cohorts. The integration of pathological markers (ER, PR, and HER2) with multi-parametric MRI radiomics enhanced performance, resulting in AUCs of 0.79, 0.71, and 0.80.

Similarly, Syed et al. [134] combined non-imaging features (pathological markers) with GLCM features to predict treatment response. However, radiomics features were extracted from DWI and DCE sequences at three distinct time points (pre-treatment, early treatment at 3 weeks, and mid-treatment at 12 weeks of NACT). Utilizing an XGBoost ML classifier with 5-fold cross-validation on publicly available data, the author’s predicted response. The mean AUCs of prediction models based on ADC, DWI, DCE, the combination of DWI and DCE, and all MRI features combined were 0.85, 0.871, 0.903, 0.916, and 0.933, respectively. Additionally, non-imaging features achieved an AUC of 0.919. The amalgamation of all MRI features with non-imaging attributes yielded optimal prediction performance, resulting in an AUC of 0.951, precision of 0.815, and recall of 0.926. The study concluded that XGBoost holds promise for early patient response prediction based on GLCM and pathological features.

Chen et al. [135] extracted a total of 984 radiomics features from pre-treatment DCE, T2WI, and DWI images. Among these, six features were selected—three from T2WI and three from DCE—to construct the radiomics signature using a random forest approach. The resulting radiomics signatures demonstrated notable performance metrics in the test set, including an AUC of 0.834, a sensitivity of 80%, and a specificity of 73.21%. The study also explored the relationship between pathological markers and pCR, revealing that ER and PR statuses could serve as independent predictors of pCR. Furthermore, a predictive nomogram was developed by incorporating ER, PR, and the radiomics signature, yielding an improved AUC of 0.879, a sensitivity of 83.57%, and a specificity of 82.19%. The combined use of DWI and T2WI, along with ER and PR status, exhibited strong predictive capabilities.

In a related work, Chen et al. [136] constructed a novel nomogram that integrated baseline radiomics features from DCE and ADC maps with pathological markers to distinguish between pCR (grade 5) and pPR (grades 1–4 on the Miller-Payne grading system). The authors employed logistic regression to establish prediction models based on ADC maps alone, DCE images alone, and a combination of ADC maps and DCE. The corresponding AUCs were measured at 0.639, 0.789, and 0.68, respectively. Additionally, an exploration of the predictive capacity of pathological markers resulted in an AUC of 0.793. Notably, the study concluded that the most effective predictive performance was achieved by combining ER and PR statuses with radiomics features from both ADC maps and DCE images. This combined approach outperformed the use of individual feature sets, achieving an AUC of 0.837, an accuracy of 0.893, a sensitivity of 0.714, and a specificity of 0.952.

Xiong et al. [137] assessed the role of T2WI, DCE, and DWI sequences in predicting the response of NACT-insensitive breast cancers prior to treatment (specifically, grade 1 & 2 cases in the Miller-Payne grading system). They constructed a prediction model that integrated four radiomics markers with two independent pathological factors (HER2 status and Ki-67 index), identified through multivariable logistic regression. This composite model achieved an AUC of 0.935 and an accuracy of 93.55%, surpassing both radiomics-only models (AUC: 0.83) and pathological marker-based models (AUC: 0.792) during validation. These findings suggest the efficacy of a nomogram based on radiomics extracted from pre-treatment multi-sequence MRI scans, along with HER2 status and Ki-67 index, in effectively predicting NACT-insensitive cases.

Joo et al. [138] developed a multimodal DL model for predicting pCR based on pre-treatment T2WI and DCE-T1WI subtraction images, along with clinico-pathological markers. They constructed two 3D ResNet-50-based CNNs to extract radiomics features from the entire bilateral MRI images. The AUCs for the models using T2WI images, DCE-T1WI subtraction images, and clinico-pathological markers were 0.663, 0.725, and 0.827, respectively. Combining T2W and DCE-T1W subtraction images produced an AUC of 0.745 and an accuracy of 73.6% (sensitivity: 48.6%, specificity: 85.3%). Furthermore, They found that the optimal prediction performance emerged from the integration of clinicopathological markers, T2W, and DCE-T1W subtraction images, yielding an AUC of 0.888, accuracy of 85%, sensitivity of 66.7%, and specificity of 93.2%.

Yoon et al. [139] utilized multiple logistic regression to assess the value of textural features extracted from baseline PET/CT and DWI scans in distinguishing responders from non-responders. The ranges of *p*-values for the selected texture features are presented in Table 5. They concluded that textural features hold predictive potential, offering insights into tumor heterogeneity.

On the other hand, Umutlu et al. [140] examined the potential of radiomics features to predict treatment response using baseline PET/MRI scans. Employing SVM with 5-fold cross-validation, they predicted pCR across the entire patient cohort and within two molecular subtype subgroups ((HR+/HER2-) and (TN/HER2+)). The combined application of all MRI sequences and PET yielded the most robust predictive performance for the entire cohort, resulting in an AUC of 0.8, an accuracy of 77.4%, a sensitivity of 81%, a specificity of 73.8%, a PPV of 75.6%, and NPV of 79.5%. Within the (HR+/HER2-) subgroup, the same imaging dataset achieved superior AUC and accuracy values (0.94 and 85.2%, respectively), accompanied by an 85.2% sensitivity and specificity. The authors’ conclusion emphasized the capacity of PET/MRI to offer an accurate radiomics analysis for early pCR prediction, particularly in HR+/HER2- patients.

A comparison was made by Choi and colleagues [141] between an Alexnet-based CNN model and conventional methods, to predict responders and non-responders to NACT based on PET/CT and DWI scans. The proposed CNN model demonstrated AUCs of 0.886, 0.98, 0.602, and 0.701, along with accuracies of 97%, 95%, 85%, and 88% for PET0, PET1, MRI0, and MRI1, respectively, (where 0 represents baseline scans and 1 represents scans after the first NACT cycle). Conversely, the AUCs of conventional parameters including SUV, TLG, MTV, and ADC remained below 0.7 in both baseline and follow-up scans, with accuracies falling short of 84%. Furthermore, improvements were observed in AUC values (0.805, 0.737, 0.758, and 0.752) for differences in these parameters between the two scans, denoted as ΔSUV_max_, ΔMTV, ΔTLG, and ΔADC.

Montemezzi et al. [142] integrated pathological and radiomics features obtained from pre-NACT PET/CT and DCE-MRI scans to predict pCR.They employed three classifiers (Random Forest, Support Vector Machine Regression (SVR), and Logistic Regression) in conjunction with leave-one-out and leave-two-out cross-validation techniques. The evaluation encompassed five feature groups (mentioned underneath Table 5). They found that the incorporation of both radiomics and pathological markers resulted in elevated AUC values of 0.96 and 0.98. The study demonstrated that the fusion of DCE-MRI and PET/CT radiomics with pathological markers led to enhanced predictive performance.

**Table 5 cancers-15-05288-t005:** Multi-modal Imaging.

Reference	Study Aim	Number of Patients & Study Type	Markers	Results & Findings
Liang et al. [126]	To investigate the usefulness of combining DCE-MRI parameters with ADC values for the early prediction of pCR to NACT.	119 patients Study type: single-center retrospective study	Clinical: N/A Pathological: N/A Radiomics: K^trans^, K_ep_, V_e_, the initial area under the curve (IAUC), washin, washout, TTP, and ADC values before and after the second cycle of NACT.	The AUCs of ADC, TTP, K_ep_, K^trans^, IAUC, and washing after the 2nd cycle were 0.721, 0.725, 0.805, 0.825, 0.824, and 0.866, respectively, while it did not exceed 0.57 for V_e_ and washout. The SENs were 87.5%, 100%, 62.5%, 83.33%, 83.33%, 83.33%, 45.83%, and 62.5%. The SPEs were 56.84%, 42.11%, 92.63%, 75.79%, 78.95%, 84.21%, 71.58%, and 58.95%. Combining ADC, TTP, & washing achieved AUC: 0.886, SEN: 87.5%, SPE: 82.11%. They found that baseline features did not show a significant difference between pCR & non-pCR; however, they showed good predictive performance after two cycles.
Li et al. [127]	To evaluate the utility of multiparametric MRI parameters acquired from DCE-MRI and DWI acquired before and after the first cycle of NACT in predicting pCR in patients with breast cancer.	42 patients (data after the first cycle of NACT was available for only 36 patients) Study type: prospective study	Clinical: N/A Pathological: N/A Radiomics: longest dimension (LD), K^trans^, K_ep_, V_e_,V_p_, ADC, and the derived parameter K_ep_/ADC. They were acquired before and after the first cycle.	The AUCs (after the first cycle of NACT) for LD, V_e_, V_p_, K^trans^, K_ep_, ADC, and K_ep_/ADC were 0.57, 0.54, 0.61, 0.68, 0.76, 0.82, and 0.88, respectively. Moreover, the SENs were 0.83, 0.67, 0.5, 0.67, 0.83, 0.83, and 0.92. The SPEs were 0.42, 0.48, 0.78, 0.74, 0.65, 0.67, and 0.78, respectively. They revealed that combining DWI & DCE parameters (i.e., K_ep_/ADC) yielded a superior performance than using each of them alone. In addition, the mean parameters after one cycle of therapy outperformed the baseline parameters and the percentage change between the two scans.
O’Flynn et al. [128]	To determine whether individual functional MRI parameters can predict pCR to NACT in breast cancer patients after two treatment cycles.	32 patients Study type: single-center prospective study	Clinical: N/A Pathological: N/A Radiomics: tumor volume, K^trans^, K_ep_, V_e_, IAUGC (IAUGC: initial area under the gadolinium curve), enhancement fraction (EF), ADC, R2* values, and their percentage changes after the second cycle of NACT.	The AUCs of the percentage change in EF, tumor volume, IAUGC, K^trans^, K_ep_, V_e_, ADC, & R2* were 0.76, 0.77, 0.64, 0.6, 0.68, 0.58, 0.69, & 0.41, respectively. SENs: 63.2%, 71.4%, 73.7%, 63.2%, 63.2%, 57.9%, 78.9%, & 63.2%. SPEs: 76.9%, 76.9%, 61.5%, 53.8%, 69.2%, 53.8%, 69.2%, & 30.8%. They found that the reduction in ER & tumor volume was significantly greater in patients achieving pCR, and they can be used as early response predictors.
Zhao et al. [129]	To investigate the ability of DWI combined DCE-MRI in the prediction of pCR after the second cycle of NACT by developing a nomogram based on MRI features.	87 patients Training (n = 66) Validation (n = 21) Study type: single-center retrospective study	Clinical: N/A Pathological: N/A Radiomics: tumor longest diameter, ADC values, type of TIC (time signal intensity curve type: persistent, plateau, and washout), maximal enhanced ratio (E_max_), early enhanced ratio (E_90_), and percentage change in these parameters.	Multivariate logistic regression showed that the following features were independent pCR predictors: ADC value after 2 cycles (OR: 1.041, *p* = 0.037) The percentage decrease in: E_90_ (OR: 0.927, *p* = 0.002) & tumor size (OR: 0.948, *p* = 0.011) The prediction model yielded AUC: 0.94, SEN: 100%, SPE: 86.7% They found no significant difference in pathological markers, age, and baseline radiomics features between the pCR and non-pCR groups. Moreover, the nomogram & the predictive model showed strong predictive value.
Bian et al. [130]	To evaluate the ability of radiomics signatures to predict the efficacy of NACT and the probability of pCR based on pretreatment T2WI, DWI, and DCE MRI scans.	152 patients Training (n = 107) Validation (n = 45) Study type: single-center retrospective study	Clinical: N/A Pathological: N/A Radiomics: Model(1) tumor size and morphology, number of tumors, the diameter of LN, and time-intensity curve. Model(2) textural features (GLCM and GRLM) extracted from pre-treatment scans.	For predicting NACT sensitivity Model(1) AUC: 0.56, SEN: -, SPE: - Model(2) AUC: 0.91, ACC: 81.8%, SEN: 100%, SPE: 75% Nomogram AUC: 0.93, ACC: 81.8%, SEN: 100%, SPE: 75% For predicting pCR to NACT Model(1) AUC: 0.64, SEN: -, SPE: - Model(2) AUC: 0.91, ACC: 88.9%, SEN: 88.2%, SPE: 90.9% Nomogram AUC: 0.91, ACC: 88.9%, SEN: 88.2%, SPE: 90.9% The authors identified the LN minimum diameter, speculated tumor margin, and tumor maximum diameter as independent predictors of response (model 1). They also revealed that model 2 and the nomogram (combining features used in model 1 and model 2) achieved similar discrimination power.
Tahmassebi et al. [131]	To assess the utility of ML algorithms in the early prediction of survival outcomes and pCR to NACT using multi-parametric MRI scans acquired before and after two cycles of NACT.	38 patients (4-fold cross-validation) Study type: single-center prospective study	Clinical: N/A Pathological: N/A Radiomics: tumor size, shrinkage pattern, [DCE: shape, margins, symmetry, internal enhancement characteristics, mean plasma flow, mean transit time (MTT), volume distribution], [T2WI: signal intensity, peritumoral edema], [DWI: max, min, & mean ADC].	The best AUCs in predicting RCB using 8 ML classifiers (LR, SVM, SGD, LDA, RF, DT, AdaBoost, and XGBoost) were 0.868, 0.88, 0.83, 0.75, 0.89, 0.81, 0.85, and 0.94, respectively, (SEN: -, SPE: -). They concluded that the XGBoost outperformed other classifiers as it achieved higher accuracy and more stable performance. Moreover, peritumoral edema, min ADC, complete shrinkage pattern, changes in tumor size, and MTT can be used as RCB predictors.
Eun et al. [132]	To determine whether texture features from different MRI sequences at pre- and mid-treatment are associated with pCR to NACT.	136 patients (5-fold cross-validation) Study type: single-center retrospictive study	Clinical: N/A Pathological: N/A Radiomics: histogram texture features (standard deviation, mean pixel intensity, mean proportion of positive pixels, entropy, skewness, & kurtosis).	Texture features at mid-treatment contrast-enhanced-T1WI showed the best performance compared to other MRI sequences, the prediction model based on random forest classifier achieved: AUC: 0.82, ACC: 83.1%, SEN: 62.5%, SPE: 91.7%. They revealed that the RF model had better performance showing the association between texture features and pCR compared with the other six ML classifiers.
Liu et al. [133]	To investigate the efficacy of radiomics analysis of pretreatment multi-parametric MRI (T2WI, DWI, and T1 + C) scans in the prediction of pCR to NACT.	414 patients Training (n = 128) Three independent validation cohorts (n = 99, 107, & 80) Study type: multi-center retrospective study	Clinical: N/A Demographic: age Pathological: ER, PR, HER2, tumor stage and Ki67 index Radiomics: shape-and-size features, textural, first-order statistical, and wavelet features (Gabor-bank & Law’s filters).	The AUCs of the three validation cohorts using: Pathological model: 0.76, 0.60, & 0.79 (SEN: -, SPE: -) Radiomics of multi-sequence MRI: 0.70, 0.68, & 0.79 (SEN: -, SPE: -) Incorporating multi-sequence radiomics & pathological markers: 0.79, 0.71, & 0.80 (SEN: -, SPE: -) They found that integrating pathological markers with radiomics extracted from multi-sequence MRI outperformed all the single-sequence radiomics and could be helpful for the pretreatment prediction of pCR. Moreover, multi-parametric MRI can provide concurrent insights into tumor morphology, micro-vascular permeability, water diffusion properties, and cellularity.
Syed et al. [134]	To integrate radiomics features extracted from DWI and DCE MRI scans with non-imaging features to predict pCR to NACT using XGBoost ML classifier. Radiomics features were extracted from pre-, early-, & mid-treatment scans	117 patients from the Breast Multi-parametric MRI for prediction of NAC Response-2 competition dataset (BMMR2) (a subset of I-SPY-2 TRIAL) competition dataset (5-fold cross-validation) Study type: retrospective study based on multi-center dataset	Clinical: N/A Demographic: age & race Pathological: lesion type, HR, HER2, and Scarff- Bloom-Richardson grade (SBR grade). Radiomics: ADC values and GLCM textural features of DWI and DCE (energy, correlation, homogeneity, dissimilarity, contrast, dissimilarity, and Angular Second Moment (ASM).	Below, the mean AUCs, SENs, SPEs, & precisions of the XGBoost prediction models based on: ADC: 0.85, 0.827, -, & 0.752. DWI: 0.871, 0.926, -, & 0.779. DCE: 0.903, 0.939, -, & 0.856. DWI + DCE: 0.916, 0.915, -, & 0.779. all MRI sequences: 0.933, 0.889, -, & 0.824. pathological markers: 0.919, 0.914, -, & 0.762. combining pathological markers with all MRI sequences: 0.951, 0.926, -, & 0.815. They found that XGBoost can accurately predict response based on non-imaging and GLCM features.
Chen et al. [135]	Predicting the efficacy of NACT by constructing a nomogram based on pathological factors and multi-sequence MRI (T2WI, DWI, and DCE).	158 patients Training (n = 110) Test (n = 48) Study type: single-center retrospective study	Clinical: N/A Demographic: age & gender Pathological: ER, PR, HER2, Ki67, & clinical stage. Radiomics: histogram parameters, GLCM, GLRLM, and form factor parameters (which are descriptors of the 3D size and shape features of the tumor).	The prediction performance of: Radiomics signature AUC: 0.834, SEN: 80%, SPE: 73.21% The nomogram (integrating radiomics signature with PR and ER status) AUC: 0.879, SEN: 83.57%, SPE: 82.19 % They revealed that ER and PR status showed significant differences between the two response groups while the other pathological markers did not. Moreover, DWI & T2WI could predict response effectively.
Chen et al. [136]	To develop a radiomics nomogram combining pre-treatment DCE-MRI and ADC maps with pathological risk factors to predict pCR to NACT.	91 patients Training (n = 63) Test (n = 28) Study type: single-center retrospective study	Clinical: N/A Demographic: age Pathological: Ki67, ER, PR, and HER2 status. Radiomics: histogram parameters, morphological features, GLCM, GLRLM, and GLSZM features extracted from ADC maps and DCE images at pre-treatment stage.	Multivariate logistic regression model yielded the following AUCs, ACCs, SENs, & SPEs in the test set when using: DCE features alone: 0.789, 78.6%, 71.4%, & 81% ADC features alone: 0.639, 53.6%, 100%, & 38.1% Combining DCE & ADC: 0.68, 78.6%, 71.4%, & 81% Pathological markers: 0.793, 75%, 57.1%, & 81% Combining pathological with radiomics (DCE+ADC): 0.837, 89.3%, 71.4%, & 95.2% They concluded that ER & PR showed a significant difference between pCR & pPR groups (*p* < 0.05). Moreover, combining radiomics from DCE-MRI & ADC maps with pathological data can be potential response predictors.
Xiong et al. [137]	To assess the value of multi-parametric MRI (T2WI, DCE, and DWI) in the prediction of NACT-insensitive breast cancers based on pretreatment scans.	125 patients Training (n = 63) Validation (n = 62) Study type: single-center retrospective study	Clinical: N/A Demographic: age Pathological: ER, PR, HER2, tumor stage and Ki67 index Radiomics: shape, first-order statistic, GLCM, GLRLM, NGTDM, GLSZM, and wavelet features.	The prediction model based on pathological markers achieved AUC: 0.792, ACC: 87.1%, SEN: -, SPE: - The model based on radiomics markers attained AUC: 0.83, ACC: -, SEN: -, SPE: - Combining radiomics with pathological markers achieved AUC: 0.935, ACC: 93.55%, SEN: -, SPE: - They suggested that a nomogram built based on HER2 status, Ki-67 index, and radiomics features extracted from pretreatment multi-parametric MRI can predict NACT-insensitive effectively.
Joo et al. [138]	To conduct a multimodal DL model that combines clinicopathological information with MR images acquired before the initiation of NACT to help in the prediction of pCR.	536 patients Training (n = 429) Validation (n = 107) Study type: single-center retrospective study	Clinical: CA 15-3 Demographic: age Pathological: body mass index (BMI), menopausal status, histologic subtypes, T stage, N stage, ER, PR, HER2, Ki-67, and NACT regimen. Radiomics: features were extracted from T2W and DCE-T1W subtraction images by two 3D CNNs based on 3D-ResNet-50.	The models based on T1W, T2W images, and clinicopathological markers individually achieved AUCs: 0.725, 0.663, and 0.827, respectively. ACCs: 71.8%, 70.9%, & 78.5%. SENs: 31.4%, 45.7%, & 84.8%. SPEs: 90.7%, 82.4%, & 75.7%. Integrating the aforementioned markers achieved AUC: 0.888, ACC: 85%, SEN: 66.7%, SPE: 93.2% They concluded that the best prediction performance was attained by combining baseline MRI images with clinical and pathological markers.
Yoon et al. [139]	To evaluate the efficacy of textural features extracted from pretreatment F-18 FDG PET/CT and DWI scans to predict the LABC patients’ pathological response to NACT and progression-free survival (PFS).	83 patients with locally advanced breast cancer (LABC). Study type: retrospective study	Clinical: N/A Pathological: N/A Radiomics: From PET/CT: TLG, MTV, SUV_max_, first-order histogram analysis of SUV, and second-order textural features (CM, NIDM, VAM, NGLCM, ISZM, NGLDM, and TSM) ([CM: co-occurrence matrix, NIDM: neighborhood intensity difference matrix, VAM: voxel-alignment matrix, NGLCM: normalized gray-level co-occurrence matrix, ISZM: intensity size-zone matrix, NGLDM: neighborhood gray-level dependence matrix, and TSM: texture spectrum matrix]). From DWI: ADC, histogram analysis, and second-order texture features.	-The range of *p*-values of the selected PET features: CM: 0.008–0.02, VAM: 0.008–0.047 NIDM: 0.012–0.024, ISZM: 0.005–0.032 NGLCM: 0.045, TSM: 0.009 NGLDM: 0.009–0.019 (SEN: -,SPE: -). -The range of *p*-values of the selected ADC features: histogram analysis (entropy): 0.024 NGLCM: 0.033–0.025 (SEN: -,SPE: -). They found that tumor texture features are useful for the prediction of NACT response in as they indicate tumor heterogeneity.
Umutlu et al. [140]	To evaluate the potential of radiomics analysis of multi-parametric ^18^F-FDG PET/MRI pre-treatment images to predict pCR to NACT.	73 patients (5-fold cross-validation) Study type: retrospective study	Clinical: N/A Pathological: N/A Radiomics: first-order statistics, GLCM, GLRLM, GLSZM, GLDM, and NGTDM extracted from pretreatment PET, ADC, DCE-MRI, and T2WI scans.	Combining the PET data with all MRI sequences achieved the following results using: Whole patients cohort AUC: 0.8, ACC: 77.4%, SEN: 81%, SPE: 73.8%, PPV: 75.6%, NPV: 79.5%. HR+/HER2- subgroup AUC: 0.94, ACC: 85.2%, SEN: 85.2%, SPE: 85.2%, PPV: 85.2%, NPV: 85.2%. TN/HER2+ subgroup AUC: 0.92, ACC: 87.5%, SEN: 88.2%, SPE: 86.7%, PPV: 88.2%, NPV: 86.7% They revealed that PET/MRI has a promising role in predicting pCR prior to treatment, especially for (HR+/HER2-) tumors.
Choi et al. [141]	To assess the value of PET/CT and DWI parameters in predicting pathological response to NACT using CNNs and compare their performance with conventional imaging parameters.	56 patients (3-fold cross-validation) Study type: single-center retrospective study	Clinical: N/A Pathological: N/A Radiomics: - Conventional parameters: SUV_max_, MTV, TLG, ADC, and their differences (Δ) between pre-NACT and after the first cycle scans. - CNN model: based on (Alexnet) and the inputs were images acquired before (PET0, MRI0) and after the first cycle (PET1, MRI1).	Using conventional parameters: The AUCs of SUV, TLG, MTV, and ADC parameters did not exceed 0.7 in baseline and follow-up scans. The AUCs for ΔSUV_max_, ΔMTV, ΔTLG, & ΔADC were 0.805, 0.737, 0.758, & 0.752, respectively. SENs: 83%, 67%, 67%, & 83%. SPEs: 68%, 80%, 80%, & 72%. Using CNN model: PET0, PET1, MRI0, & MRI1, respectively, achieved AUCs of 0.886, 0.98, 0.602, & 0.701. ACCs: 97%, 95%, 85%, & 88%. SENs: 79%, 72%, 18%, & 14%. SPEs: 94%, 96%, 90%, & 90%. They found that ΔSUV_max_ achieved the highest prediction performance. Using CNNs instead of conventional methods improved the prediction performance for all parameters except ADC.
Montemezzi et al. [142]	To study the effect of combining DCE-MRI radiomics with histological and radiological information (PET/CT) on the performance of the prediction model of pCR to NACT.	60 patients (leave one out & leave two out cross-validation 60-fold and 30-fold cross-validation). Study type: single-center retrospective study	Clinical: N/A Demographic: Age Pathological: tumor type and grade, ER, PR, HER2, and Ki-67. Radiomics: tumor shape, margin, internal enhancement, type of enhancement curve, zero (geometric), first, and higher order (textural features).In addition to ADC (from DWI) and SUV_max_(from PET/CT).	LR, SVR, and RF were used with different combinations of markers (5 groups of features (The five groups of features were: (1) tumor characteristics such as shape, type, grade, margin, internal enhancement, curve type, SUV_max_, ADC, and patient age; (2) the selected radiomics features; (3) pathological features; (4) a combination of features from groups 1 and 2; and (5) an amalgamation of features from groups 1, 2, and 3.)) , and the AUCs were as follows: (SENs: -, SPEs: -) Group 1: 0.7–0.75 Group 3 (pathological): 0.8–0.85 Group 4: 0.85–0.9 Group 5: 0.96–0.98 (using LR) They found that the introduction of DCE-MRI radiomics showed significant improvement in predictive power. The selected radiomics were dependence variance, sphericity, kurtosis, and LRHGLE (LRHGLE: long run high gray-level emphasis).

## 7. Discussion

This comprehensive review article has delved into the intricate challenge of the variable responses observed in NACT. The central thrust of this discourse lies in the exploration and development of predictive models capable of stratifying patients based on their probable response to NACT. A nuanced emphasis has been placed on the evolution of the predictive landscape, specifically the integration of radiomic and pathological markers. The objectives sought to uncover resilient radiomic markers that exhibit correlations with NACT response, while concurrently probing the potential amplification of predictive precision through the amalgamation of radiomic markers derived from radiological images and classical pathological markers.

The endeavor to unravel the complexities of NACT response prediction has invoked diverse avenues of research. Extensive investigations have been conducted, traversing the terrain of the tumor core and its margins (peritumoral regions), as evident in [58,98,107]. A pivotal narrative echoed by [46,108,113,120] accentuates the merits of encompassing a broader spatial context within feature extraction strategies.

Our review has shown that the predictive landscape of NACT responses has undergone considerable evolution, especially with the incorporation of both radiomic and pathological markers. Emphasis on the evolution of this landscape becomes apparent when noting that 96% of articles published in the last 15 years employ radiomic markers derived from radiological images. Furthermore, 92% of these articles have documented a statistically significant correlation between at least one radiomic feature and NACT response.

Diverse imaging modalities have been harnessed to navigate the objective of NACT outcomes prediction. **Mammography**,traditionally seen as the cornerstone of breast tumor detection, has undergone a transformative phase with artificial intelligence integrations. Various features have been extracted from mammogram images to predict the outcome of NACT. Shape features, textural features, and the percentage of gray value change between scans have been utilized as predictive indicators. Shin et al. [35] and Skarping et al. [36], for instance, have effectively used AI to derive insights from mammogram images, with Skarping et al. distinguishing between pCR and non-pCR, recording a specificity of 90% and sensitivity of 46%. Moreover, CESM, as demonstrated by Xing et al. [44], has struck a balanced performance with a specificity of 72.15% in the CC view and 75% sensitivity. The research landscape, marked by studies like those from Wang et al. [45] and Mao et al. [46], is hinting toward the increasing importance of radiomics, with sensitivities and specificities touching the ranges of 57.7% and 90.9% respectively.

A parallel trajectory unfolds in studies utilizing **ultrasound**, where QUS parameters (SS, SI, MBF, ASD, AAC, ACE, and SAS) have been scrutinized extensively, often integrated with textural analyses. The exploration of elastography variants, namely strain elastography and shear-wave elastography, has also unveiled pivotal features such as strain ratio, tumor stiffness, and elasticity. Noteworthy is the DL radiomics nomogram by Jiang et al. [50], which registers a sensitivity of 89.33%, and a specificity of 81.37%.

The scope broadens further, encapsulating **PET/CT** investigations, where ΔSUV_max_ represents the maximum standardized uptake value change between baseline and follow-up scans. Augmenting this are MTV, TLG, SUV_peak_, SUV_mean_, and an array of textural features, contributing collectively to the predictive tapestry. Research spearheaded by Buchbender et al. [80], Andrade et al. [75], and Groheux et al. [82,83] have established ΔSUV_max_’s promise, with sensitivities ranging from 75% to 85.7%. Moreover, merging ΔSUV_max_ with textural features, as demonstrated by Cheng et al. [86], holds promise in enhancing predictive accuracy through a multi-faceted approach.

However, amidst all imaging modalities, **DCE-MRI** emerges as the undeniable frontrunner. Its omnipresence in the literature and the stellar results linked with it accentuate its unparalleled value. The domain of DCE-MRI has proven equally intricate, unfurling a plethora of features. These encompass quantitative parameters like K^trans^, K_ep_, V_e_, V_p_, and τ_*i*_. Semi-quantitative parameters like wash-in, wash-out, and peak enhancement intertwine with morphological, statistical, and textural features such as GLCM and GLRLM. Moreover, the convergence of DCE-MRI with DWI holds promise, prominently featuring ADC maps and values (maximum, minimum, and mean) as pivotal predictors. The versatility of DCE-MRI allows for the integration of data from digital breast tomosynthesis. Further, its adeptness in synthesizing molecular subtypes, clinical markers, and radiomic features, has a capability that appears challenging for other modalities to rival. For instance, Comes et al. [123]’s combined strategy (incorporating pathological features with radiomics features) achieved an impressive AUC of 0.9 was achieved, along with an accuracy of 92.3%, a sensitivity of 85.7%, and a specificity of 94.7%. Additionally, Hussain et al. [120] reached an AUC of 0.98 and an accuracy of 94%, emphasizing the potential of DCE-MRI’s radiomics when integrated with the molecular subtype. Nevertheless, it’s imperative to emphasize that while DCE-MRI seems superior, the best modality or marker may still oscillate depending on the clinical scenario, resource allocation, and individualized patient parameters.

Shining brightly on the horizon is the concept of **multi-modal imaging** that truly provides insights into the potential of integrating predictive markers from various imaging modalities. As this approach beckons exploration, it potentially paves the way for even more refined and accurate predictions in the near future. Across the evaluated imaging modalities, certain markers stood out as robust indicators of treatment response. Notably, in MRI studies, texture features from contrast-enhanced T1-weighted images at mid-treatment consistently exhibited strong predictive power. Noteworthy PET parameters, such as TLG and ΔSUV_max_, also showed substantial promise in prognosticating pCR. Variables such as hormone receptor status (ER/PR), HER2 expression, and Ki-67 proliferation index were frequently linked to treatment outcomes. Integrating these factors with multi-modal imaging data, such as radiomics extracted from PET/CT or MRI, markedly improved predictive accuracy. Joo et al. [138], a prime example, showcased a multimodal DL model that combined clinicopathological information with pre-treatment MRI images, attaining a high specificity of 93.2%, an accuracy of 95% and an AUC of 0.89 for post-first-cycle PET images. Similarly, Umutlu et al. [140] leveraged multiparametric PET/MRI to predict pCR with promising results (average AUC = 0.93, 86.4% accuracy, 86.7% sensitivity, and 86% specificity). The fusion of pathological markers with radiomics features has demonstrated varying impacts on prediction models. In most cases, integrating pathological markers has led to improved prediction capabilities. However, exceptions like [45,46,56,64,65,82,116] highlight instances where pathological factors did not significantly correlate with response or failed to enhance model performance.

These clinical implications of accurate NACT response prediction are profound, offering the potential to revolutionize personalized treatment strategies and improve patient outcomes. By identifying patients who would benefit from NACT and sparing others from potentially ineffective treatment, predictive models align with the broader goals of precision medicine.

## 8. Limitations and Future Perspectives

Despite these promises of NACT response predictive models, there are clear limitations and challenges to be addressed. Differences and variations in therapeutic regimens, tumor molecular subtypes, scan timings, imaging protocols, and sample sizes have led to inconsistencies in the results across studies.

Notably, when the same predictors were employed, different studies yielded conflicting outcomes. For instance, SUV_max_ showed a correlation with pathological response in TNBC patients as indicated by Groheux et al. [83]. However, contrasting findings were presented by Luo et al. [85] and Groheux et al. [82], showing no significant correlation for LABC and HER2+ patients, respectively. Additionally, Koolen et al. [81] highlighted that ΔSUV_max_ might be a significant predictor after two weeks of NACT for TNBC patients. Yet, this predictor was not found significant in a study for HER2+ patients by Groheux et al. [82].

Regarding PK parameters extracted from DCE-MRI at baseline scans, disagreements were observed across studies. While Drisis et al. [103] found K^trans^ could differentiate responders from non-responders, Braman et al. [107] argued that PK parameters did not show any significant difference between pCR and non-pCR. Lee et al. [105] also asserted that individual perfusion parameters lacked high predictive capability. Tudorica et al. [102] and Li et al. [101] presented percentage changes in PK parameters after one NACT cycle as effective predictors, a finding that was contradicted by Cho et al. [121]. Additionally, the predictive performance evaluation results of the models trained by Fan et al. [97] on two independent patient sample groups showed some differences.

Apart from the aforementioned inconsistencies, another pertinent limitation is the largely retrospective nature of many studies, often encompassing only a small cohort of patients. These studies that might need further validation include those conducted by Sadeghi-Naini et al. [48], Tadayyon et al. [52,56], Antunovic et al. [87], Giannini et al. [96], Li et al. [101], and Crippa et al. [90]. The absence of a standardized protocol further complicates matters, with each study employing unique methodologies, features, CNN layers, and classifiers. The use of multiple software applications for radiomics feature extraction only adds to the lack of uniformity. Moreover, the high cost and complexity associated with AI applications in routine imaging require specialized software and personnel expertise.

From a forward-looking perspective, future research should aim to bridge the remaining gaps in the field. While our understanding of tumor biology deepens and AI technologies advance, the promise of informed NACT strategies to transform patient care becomes increasingly tangible. The domain is broad and rich, extending from the exploration of diverse imaging modalities to the integration of clinical and molecular data. This should be built upon a strong foundation of prospective clinical trials and expansive, balanced multi-institutional datasets. As the field evolves, advanced models can also be refined to contend with dataset limitations, yielding robust, validated, and clinically impactful predictive models.

It is crucial to prioritize the development of standardized protocols, including imaging techniques, feature extraction methodologies, and analysis frameworks. Harmonizing these protocols can reduce variations and facilitate more comparable and reproducible research outcomes. These initiatives can bolster the validity of study findings and enhance the generalizability of predictive models. Moreover, fostering collaborations across research institutions can lead to a broader consensus on predictors and their efficacies, potentially ironing out the inconsistencies that currently plague the field.

In addition to addressing inconsistencies, it’s also imperative to focus on the integration of AI technologies with traditional radiomic analyses. While the potential of AI in enhancing NACT prediction is immense, its adoption is currently hindered by several factors, including the complexities of AI software and a lack of familiarity among radiologists. Efforts should be made to demystify AI technologies and promote their wider acceptance and integration into clinical practice. Workshops, training programs, and interdisciplinary collaborations can bridge the knowledge gap and ensure that the full potential of AI-driven models is realized in the realm of breast cancer treatment. While the trajectory of breast cancer treatment is poised for transformation, the path forward requires concerted efforts to address present challenges. By resolving inconsistencies, standardizing protocols, and embracing AI advancements, we can aspire for a future where NACT response predictions are both reliable and integral to personalized breast cancer management.

## 9. Conclusions

Breast cancer remains a global challenge necessitating innovative treatments. Neoadjuvant chemotherapy (NACT) holds promise in improving outcomes by diminishing tumor sizes and facilitating less invasive surgeries. However, the diverse patient responses underscore the need for accurate predictive models. This review extensively examines NACT response prediction by integrating radiomic and pathological markers from various imaging modalities. Valuable radiomic features are identified, particularly when considering markers from both tumor core and peritumoral margins. While each imaging modality has its merits, DCE-MRI stands out when combined with pathological markers and clinical data, a sentiment echoed across various studies. Despite challenges like limited data, collaborative efforts can build robust AI models that aid evidence-based clinical decisions. Accurate NACT response prediction has profound clinical significance, optimizing outcomes and reducing unnecessary interventions. With the advancement of AI technologies and collaboration paving the way for personalized care, the synergy between radiomic and pathological markers and the potential of informed NACT strategies can revolutionize breast cancer treatment.

## Figures and Tables

**Figure 1 cancers-15-05288-f001:**
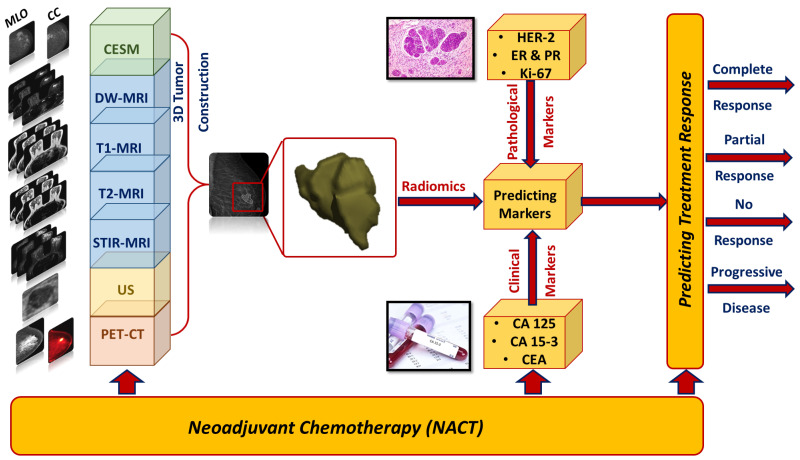
The main components of a typical AI-driven treatment response prediction system.

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
