# Peer review of "Exploring Neoadjuvant Chemotherapy, Predictive Models, Radiomic, and Pathological Markers in Breast Cancer: A Comprehensive Review"

_cancers, 2023, doi:10.3390/cancers15215288_

Round 1

Reviewer 1 Report

Comments and Suggestions for Authors

This manuscript is a very large and comprehensive review of studies aiming to predict response to neoadjuvant chemotherapy with radiomic data.

The major weaknesses of this paper are:

1/ This manuscript is very long, hard to read and without synthesis of very heterogeneous data

2/ the introduction and definition of the purpose of neoadjuvant chemotherapy is outdated. The aim of neoadjuvant chemotherapy therapy is not only reduction of volume of tumor in order to avoid indication of non-conservative surgery but mostly, now, to evaluate sensitivity of the tumor to treatment in order to adapt “rescue” treatment in the adjuvant setting. This is the current and recommended strategy by international consensus for triple negative and HER2 amplified breast cancer.

3/ In the various tables listing the studies, the main criteria of evaluation is not always defined. In some cases, pCR (pathological complete response) is given but in the majority of studies response to treatment is mentioned, without explaining if it is clinical response, pCR or radiological response

4/ Neoadjuvant treatments in breast cancer have evolved including now targeted therapy as monoclonal antibodies, immunotherapies, various chemotherapy drugs and endocrine treatments. Before evaluating probability of response according to imaging, this diversity and adaptation of methods of evaluation should be discussed.

5/There is no separate analysis of studies based on only baseline data and others studies incorporating repeated procedure during NACT, which could modify the accuracy of the methods but different cost and constraints

6/ More synthetic tables with essential information, easier to read and more useful for critical analysis would be more interesting for the reader

7/ Recent important studies are not included in this review as for example:

The accuracy of breast MRI radiomic methodologies in predicting pathological complete response to neoadjuvant chemotherapy: A systematic review and network meta-analysis.

O'Donnell JPM, Gasior SA, Davey MG, O'Malley E, Lowery AJ, McGarry J, O'Connell AM, Kerin MJ, McCarthy P.Eur J Radiol. 2022 Dec;157:110561. doi: 10.1016/j.ejrad.2022.110561. Epub 2022 Oct 17

A model combining pretreatment MRI radiomic features and tumor-infiltrating lymphocytes to predict response to neoadjuvant systemic therapy in triple-negative breast cancer.

Jimenez JE, Abdelhafez A, Mittendorf EA, Elshafeey N, Yung JP, Litton JK, Adrada BE, Candelaria RP, White J, Thompson AM, Huo L, Wei P, Tripathy D, Valero V, Yam C, Hazle JD, Moulder SL, Yang WT, Rauch GM.Eur J Radiol. 2022 Apr;149:110220. doi: 10.1016/j.ejrad.2022.110220. Epub 2022 Feb 15.

Prediction of the Pathological Response to Neoadjuvant Chemotherapy in Breast Cancer Patients With MRI-Radiomics: A Systematic Review and Meta-analysis.

Pesapane F, Agazzi GM, Rotili A, Ferrari F, Cardillo A, Penco S, Dominelli V, D'Ecclesiis O, Vignati S, Raimondi S, Bozzini A, Pizzamiglio M, Petralia G, Nicosia L, Cassano E.Curr Probl Cancer. 2022 Oct;46(5):100883. doi: 10.1016/j.currproblcancer.2022.100883. Epub 2022 Jul 21.PMID: 35914383

Machine learning with magnetic resonance imaging for prediction of response to neoadjuvant chemotherapy in breast cancer: A systematic review and meta-analysis.

Liang X, Yu X, Gao T.Eur J Radiol. 2022 May;150:110247. doi: 10.1016/j.ejrad.2022.110247. Epub 2022 Mar 10.PMID: 35290910

Reviewer 2 Report

Comments and Suggestions for Authors

Minor revisions:

1. The description of previous findings is redundant in the main text. Please be concise and conclusive;

2. Please provide the perspective opinions in this field related to breast cancer.

Reviewer 3 Report

Comments and Suggestions for Authors

Comments on this review article:

1. The overall framework is reasonable, and the content is abundant and comprehensive. The article systematically summarizes and evaluates relevant research progress in predicting neoadjuvant chemotherapy response based on radionics and pathological markers from different medical imaging modalities, demonstrating good scientific merit.

2. The article adopts an evidence-based medicine perspective, collecting and reviewing many high-quality relevant literature from 2010-2023. The content is innovative, with a particular focus on the potential of integrated multi-modality imaging applications in predicting neoadjuvant chemotherapy response.

3. The article provides a detailed summary and discussion of the efficacy of multiple predictive indicators across different imaging modalities (such as X-ray, ultrasound, PET/CT, MRI, etc.), demonstrating good comprehensiveness. At the same time, it also notes the limitations of single predictive indicators and emphasizes that combining pathological features can improve predictive accuracy.

4. The article summarizes the deficiencies of current research relatively well, such as limited sample sizes, imbalanced datasets, and the challenges of algorithm clinical application. It also points out directions for future research.

5. However, the details of individual studies could be further strengthened in the article, such as presenting the sample sizes, study types, specific definitions of predictive indicators, etc. more specifically to facilitate readers' deeper understanding and comparison of different study designs.

6. Additionally, discussions on the limitations and controversies of the research could be added to the end of the article, such as inconsistencies in results between different studies, reproducibility and stability issues with certain predictive indicators, etc. The addition of these contents could make the overall evaluation more comprehensive and rigorous.

7. Overall, this article has abundant content and a reasonable framework, providing a relatively systematic and comprehensive review of research progress in this field, with good scientific value. Further improving certain detailed descriptions and adding discussions on limitations and controversies could further enhance the academic value of the article.

Here are some examples to further describe in detail some of the deficiencies of this review article:

Sample size and study design

For example, the study by Sadeghi-Naini et al. [37] mentioned in the article included only 20 patients with locally advanced breast cancer, with a relatively small sample size. Tadayyon et al. [45] used a sample of 56 patients, which also had the problem of small sample size. The results of these single-center sample studies may be biased and have limited generalizability.

 In addition, most of the studies included in the article were retrospective cohort studies, with only a few prospective studies such as the study by Tadayyon et al. [41]. Prospective studies can better evaluate the practical value of predictive models, so the conclusions of retrospective studies still need to be treated with caution.

Definition and extraction of predictive indicators

For example, Teruel et al. [84] extracted 16 GLCM texture features, but the specific calculation method for each feature was not detailed in the article. In addition, quantitative ultrasound parameters such as SS, SI, and MBF used by Tadayyon et al. [41] also lacked specific formula descriptions. The accurate definition of these indicators is very important for explaining the results.

Inconsistent results

The evaluation of the same predictive indicators by different studies is not completely consistent. For example, Groheux et al. [71] found that SUVmax before treatment was not significantly correlated with pathological complete response in HER2-positive breast cancer patients. However, the study by Koolen et al. [70] showed that SUVmax changes after 2 weeks of treatment had predictive value in triple-negative breast cancer patients. This may be related to differences in study population composition.

Stability and reproducibility of prediction models

For example, the predictive performance evaluation results of the models trained by Fan et al. [86] on two independent patient sample groups showed some differences. This suggests that the stability and reproducibility of the model still need further validation. In addition, different optimal predictive indicators obtained for the same problem also reflect the instability of prediction results.

The above are some detailed examples of possible deficiencies of this review article, including study design, definition of predictive indicators, inconsistency of results, and model stability. Supplementing these contents will help readers understand the limitations of this research field and the problems that need to be solved more comprehensively.

Comments on the Quality of English Language

Here is the translation of your comments regarding the English language quality of the paper: Overall, the paper is well-written and uses accurate technical terminology. However, there are still some areas that need improvement: Some sentences are lengthy and have complex structures. Breaking them into shorter, simpler sentences can enhance readability and clarity. While abbreviations are defined, their usage is not consistently uniform. Abbreviations should be used consistently throughout the paper. Some of the headings have inconsistent formatting. Using a consistent heading hierarchy and format would enhance the organization. There are some spelling errors and inconsistencies in word usage that need careful proofreading. The overall tone and language of the paper align with academic standards, but in some instances, changing passive voice sentences to active-voice would be better. It's advisable to avoid excessive use of "we" and instead use terms like "this study" or "the authors" to enhance clarity. Overall, the language quality of the paper is good, with appropriate vocabulary and academic terminology. With meticulous proofreading and editing to improve grammar, sentence clarity, and format consistency, the English quality can meet publishing standards.

Round 2

Reviewer 1 Report

Comments and Suggestions for Authors

My comments were not recommendations for improvement  but examples of the major weaknesses of  the manuscript which still remain.

No organization of the review according to subtype and kind of treatment, not a comprehensive review and absence of synthesis of the data
